# Genome-wide association studies of metabolites in Finnish men identify disease-relevant loci

Xianyong Yin [1], Lap Sum Chan [1], Debraj Bose [1], Anne U. Jackson [1], Peter VandeHaar [1], Adam E. Locke [2], Christian Fuchsberger[1,3], Heather M. Stringham [1], Ryan Welch [1], Ketian Yu [1], Lilian Fernandes Silva [4], Susan K. Service[5], Daiwei Zhang [1,6], Emily C. Hector [7], Erica Young [2,8], Liron Ganel [2], Indraniel Das[2], Haley Abel[9], Michael R. Erdos [10], Lori L. Bonnycastle[10], Johanna Kuusisto[4,11], Nathan O. Stitziel [2,8,12], Ira M. Hall [13], Gregory R. Wagner[14], FinnGen*, Jian Kang[1], Jean Morrison [1], Charles F. Burant[15], Francis S. Collins[10], Samuli Ripatti [16,17,18], Aarno Palotie [16,17,19], Nelson B. Freimer [5], Karen L. Mohlke [20], Laura J. Scott [1], Xiaoquan Wen [1], Eric B. Fauman [21,22✉], Markku Laakso [4,22✉] & Michael Boehnke [1,22✉]

Few studies have explored the impact of rare variants (minor allele frequency < 1%) on highly heritable plasma metabolites identified in metabolomic screens. The Finnish population provides an ideal opportunity for such explorations, given the multiple bottlenecks and expansions that have shaped its history, and the enrichment for many otherwise rare alleles that has resulted. Here, we report genetic associations for 1391 plasma metabolites in 6136 men from the late-settlement region of Finland. We identify 303 novel association signals, more than one third at variants rare or enriched in Finns. Many of these signals identify genes not previously implicated in metabolite genome-wide association studies and suggest mechanisms for diseases and disease-related traits.

[1] Department of Biostatistics and Center for Statistical Genetics, University of Michigan School of Public Health, Ann Arbor, MI 48109, USA. [2] McDonnell Genome Institute, Washington University School of Medicine, St Louis, MO 63108, USA. [3] Institute for Biomedicine, Eurac Research, Bolzano 39100, Italy. [4] Institute of Clinical Medicine, Internal Medicine, University of Eastern Finland, Kuopio 70210, Finland. [5] Center for Neurobehavioral Genetics, Jane and Terry Semel Institute for Neuroscience and Human Behavior, University of California Los Angeles, Los Angeles, CA 90024, USA. [6] Department of Biostatistics, Epidemiology and Informatics, University of Pennsylvania Perelman School of Medicine, Philadelphia, PA 19104, USA. [7] Department of Statistics, North Carolina State University, Raleigh, NC 27695, USA. [8] Cardiovascular Division, Department of Medicine, Washington University School of Medicine, St Louis, MO 63110, USA. [9] Department of Medicine, Washington University School of Medicine, St. Louis, MO 63110, USA. [10] Molecular Genetics Section, Center for Precision Health Research, National Human Genome Research Institute, National Institutes of Health, Bethesda, MD 20892, USA. [11] Center for Medicine and Clinical Research, Kuopio University Hospital, Kuopio 70210, Finland. [12] Department of Genetics, Washington University School of Medicine, St Louis, MO 63110, USA. [13] Center for Genomic Health, Department of Genetics, Yale University, New Haven, CT 06510, USA. [14] Metabolon, Inc., Morrisville, NC 27560, USA. [15] Department of Internal Medicine, University of Michigan, Ann Arbor, MI 48109, USA. [16] Institute for Molecular Medicine Finland, FIMM, HiLIFE, University of Helsinki, Helsinki 00290, Finland. [17] Department of Public Health, University of Helsinki, Helsinki 00014, Finland. [18] Broad Institute of MIT & Harvard, Cambridge, MA 02142, USA. [19] Analytic and Translational Genetics Unit, Department of Medicine, Department of Neurology, and Department of Psychiatry, Massachusetts General Hospital, Boston, MA 02114, USA. [20] Department of Genetics, University of North Carolina at Chapel Hill, Chapel Hill, NC 27599, USA. [21] Internal Medicine Research Unit, Pfizer Worldwide Research, Development and Medical, Cambridge, MA 02139, USA. [22] These authors jointly supervised this work: Eric B. Fauman, Markku Laakso, Michael Boehnke. *A list of authors and their affiliations appears at the end of the paper. ✉email: Eric.Fauman@pfizer.com; markku.laakso@uef.fi; boehnke@umich.edu

The Finns are a geographically and linguistically isolated population who have experienced multiple population bottlenecks and expansions. This population history has resulted in large allele-frequency differences between Finns and non-Finnish Europeans (NFE), which are most pronounced in northern and eastern Finland, regions first settled in the 15th–16th centuries ("late settlement Finland")[1]. In a previous study of 64 cardiometabolic traits in ~ 20,000 individuals from these regions, we took advantage of the enrichment of otherwise rare alleles to identify 26 novel trait-associated rare deleterious alleles, 19 of which were > 20-fold more frequent in late settlement Finns than in NFE[2]. These results suggested this allele-frequency enrichment could be leveraged to identify novel rare-variant associations for additional quantitative traits. Here, we do so, reporting genome-wide association study (GWAS) results for 1391 plasma metabolites (Metabolon platform) in 6136 participants in METSIM, a study of middle-aged and older men recruited from a single site in late-settlement northeast Finland[3], who were part of our previous study[2].

Metabolites are small molecules that play a pivotal role in cellular and physiological processes and their observed levels in biofluids can reflect those processes[4]. Most metabolomics studies are performed in blood (plasma or serum) which reflects the aggregate production and consumption of metabolites by tissues[4]. Abnormal metabolite levels are commonly associated with human diseases and disease-related traits, making them useful aids to understand disease mechanisms and to identify biomarkers for disease diagnosis, prognosis, and treatment monitoring[4]. Many metabolites are highly heritable, and previous metabolite GWAS have identified common variants;[5–15] the impact of rare variants on metabolites is less well studied[16,17].

We identify 2030 independent association signals (metabolite-index variant pairs) for 803 metabolites and demonstrate 946 genetic colocalizations of 248 metabolites with 105 diseases and disease-related traits. Many of these associations identify genes not previously implicated in metabolite GWAS and suggest mechanisms for these diseases and traits. Of the 2030 association signals, 303 are novel; of these 303 signals, 111 are at 70 variants rare or > 10-fold more frequent ("enriched") in Finns compared to NFE, 78 are for 44 metabolites identified since 2015 on the Metabolon platform, and 17 are at variants on the X chromosome, which has often been ignored in previous metabolite GWAS. This study highlights the advantages of the Finnish population for rare-variant genetic association studies and the utility of integrating metabolite and disease genetic associations in disentangling disease mechanisms.

## Results

**Study design**. We assayed 1544 plasma metabolites using the Metabolon DiscoveryHD4 mass spectrometry platform (Supplementary Table 1 and Supplementary Data 1) in 6136 randomly-selected METSIM participants who were non-diabetic at baseline and passed quality control (QC) (Supplementary Table 2; Fig. 1). 1391 metabolites were successfully quantified in ≥500 of these 6136 participants. We created a METSIM imputation reference panel of > 26M genetic variants by integrating genome and exome sequence and array genotypes in 2922 METSIM participants ("Methods"; Supplementary Table 3). We used this reference panel to impute genotypes in all METSIM participants. To discover genetic mechanisms for plasma metabolite levels, we performed GWAS and statistical fine-mapping analysis and nominated putative causal genes for metabolites. We integrated metabolomics with FinnGen disease GWAS to understand disease mechanisms through genetic colocalization and Mendelian randomization analysis (Fig. 1).

**GWAS on 1391 metabolites**. We carried out GWAS across > 16M variants with imputation $r^2 \geq 0.3$ and minor allele count (MAC)≥5 in the 6136 METSIM participants for the 1391 (correlated) metabolites ("Methods"; Supplementary Table 4; Fig. 1). Single-variant association tests identified 305,555 associations at 109,368 variants for 803 metabolites at $P < 7.2 \times 10^{-11} = 5.0 \times 10^{-8}/692$ (Bonferroni correction for 692 principal components that together explained 95%[15] of phenotypic variance for the 1391 correlated metabolites; "Methods"). The GWAS $p$-values for each metabolite were well calibrated (genomic control inflation factor median = 1.00, range = 0.92–1.07; Supplementary Fig. 1). We built a multi-phenotype GWAS browser ("PheWeb") (https://pheweb.org/metsim-metab/) to visualize and make publicly available our results for all 1391 GWAS (Fig. 2; see "Discussion").

Since body mass index (BMI) influences levels of many metabolites[18], we repeated all 1391 GWAS with BMI as an additional covariate. Results with and without BMI adjustment were generally very similar, with Pearson correlation coefficient r = 0.999 for effect size estimates and $-\log_{10}p$-values for variant-metabolite pairs with $P < 7.2 \times 10^{-11}$ in either of the two analyses (Supplementary Fig. 2). Supplementary Data 2 lists the 83 associations with substantially different effect sizes (ratio ≥ 1.20) with and without BMI adjustment. In what follows, we present results for analyses without BMI adjustment.

**Detecting independent association signals**. To identify (nearly) independent association signals, we carried out chromosome-wide stepwise conditional analysis for each chromosome-metabolite pair with ≥1 association at $P < 5.0 \times 10^{-8}$. Conditional analysis identified 2030 association signals at 1143 index variants for 803 metabolites at $P < 7.2 \times 10^{-11}$ (Table 1; Supplementary Data 3; Supplementary Figs. 3–4). The 1143 index variants were of high imputation quality ($r^2$ median = 0.99, range = 0.63–1.00). 311 (27.2%) of the 1143 index variants were associated ($P < 7.2 \times 10^{-11}$) with ≥2 metabolites, suggesting widespread pleiotropy (Supplementary Fig. 5). Among the 1143 index variants, 121 (for 125 metabolites) are rare in METSIM and 99 (for 148 metabolites) have minor allele frequency (MAF) > 10-fold greater in METSIM than in NFE (gnomAD v3.1); 58 of these variants are both rare and enriched in Finns (Fig. 3a; Supplementary Data 3).

Index variants explained from 0.7% to 62.0% (median = 1.4%; Supplementary Fig. 6) of the phenotypic variance of the corresponding metabolite; 99 index variants explained ≥10% of the variance (Supplementary Data 4), including three missense variants with > 10-fold greater frequency in METSIM than in NFE. For example, the putatively-deleterious *AFMID* missense variant p.Ala41Pro (rs77585764; MAF = 5.4% in METSIM vs. 0.38% in NFE) explained 15.5% of the variance in N-formylanthranilic acid. *AFMID* encodes arylformamidase, an enzyme that catalyzes N-formylanthranilate to produce anthranilate and formate[19].

**Fine mapping**. To fine map the causal variants for the 2030 association signals, we created 2 Mb regions centered on each index variant and merged overlapping regions associated with the same metabolite, resulting in 1501 regions. We used Bayesian fine-mapping[20] with a uniform prior to calculate the variant posterior inclusion probability (VPIP) that each variant is causal and the signal posterior inclusion probability (SPIP), the sum of the VPIPs for the variants in a region ("Methods"). This method can identify multiple independent signals in a region. In the 1501 regions, we identified 2435 signals with SPIP ≥ 0.95, 1952 of which are among the 2030 association signals identified in conditional analysis. For these 1952 signals, we built 95% credible

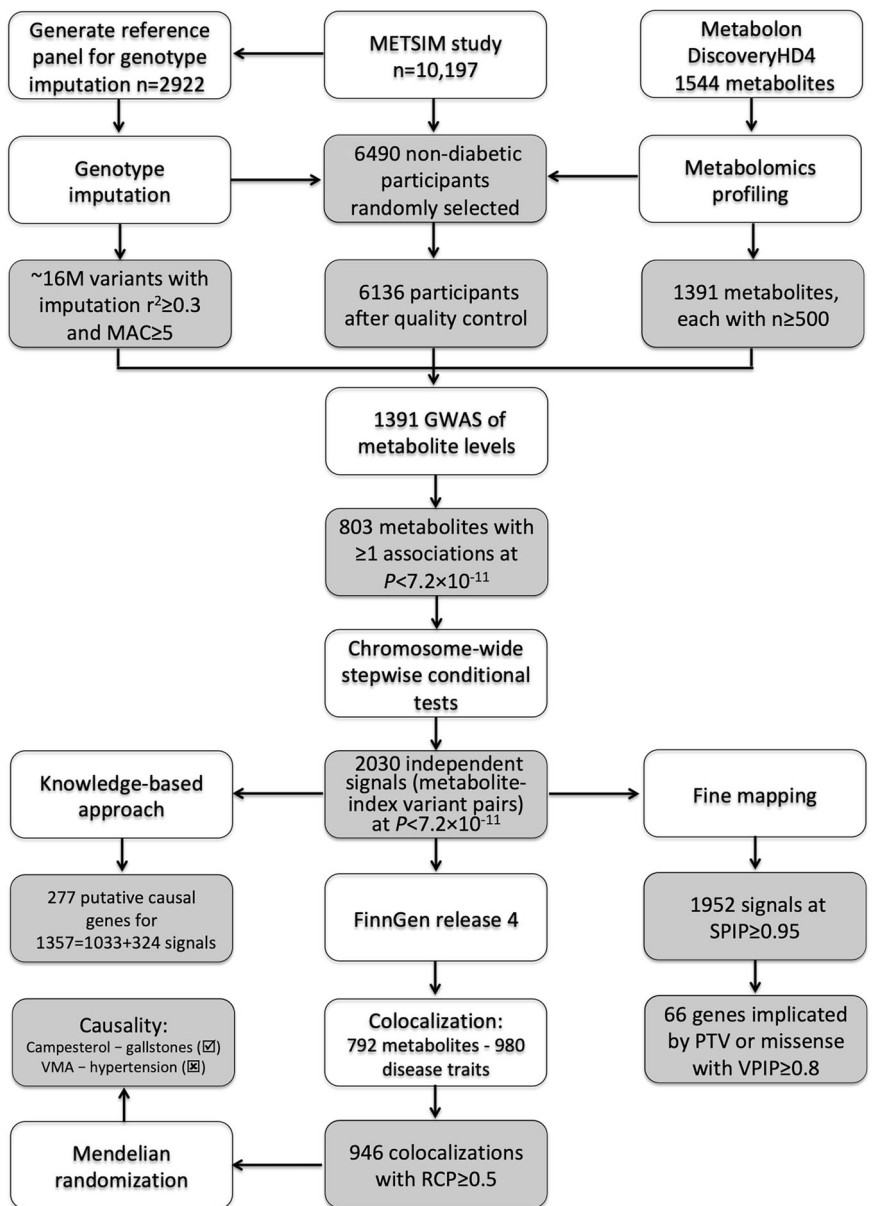

**Fig. 1 Flow chart of the METSIM metabolomics study.** MAC: minor allele count; SPIP and VPIP: signal and variant posterior inclusion probability in DAP-g Bayesian fine mapping; RCP: regional colocalization posterior probability in FastENLOC; PTV: protein-truncating variant; VMA: vanillylmandelate.

sets[21] of potential causal variants, the minimal subset of variants with summed VPIP ≥ 0.95 (Fig. 4a; Supplementary Data 5). These credible sets included 1–544 variants (median = 6); notably, 334 credible sets included only one variant (168 distinct variants).

In each of the 1952 credible sets, we identified the variant with the largest VPIP. This list comprised 1119 distinct variants, 100 with MAF > 10-fold greater in METSIM than in NFE. VPIPs for these 100 variants were greater than those for the remaining 1019 (VPIP mean = 0.73 vs. 0.47; $t$-test $P = 1.7 \times 10^{-21}$; Fig. 4b). Among the 1119 variants, 150 were shared between two signals and 146 by ≥ 3 (up to 39).

Of the 1119 variants, 263 had VPIP ≥ 0.8 in 547 credible sets. Among these 263 variants, 46 are rare in METSIM and 47 have MAF > 10-fold greater in METSIM than in NFE; 28 of these variants are both rare and enriched in Finns (Supplementary Data 5). The 263 variants include 11 protein-truncating (PTV) and 69 missense variants across 66 genes, and 183 other (mostly non-coding) variants (Supplementary Data 5). Given their likely impact on gene function, we focused on the 80 = 11 + 69 PTV

and missense variants, which suggested causal roles for the corresponding 66 genes. These 80 variants had VPIP ≥ 0.8 in credible sets for 208 signals with 173 metabolites. Among the 80 variants, 26 (5 PTV and 21 missense) are rare and 30 (6 PTV and 24 missense) have MAF > 10-fold greater in METSIM than in NFE; 16 of these variants are both rare and enriched in Finns.

**Identifying novel association signals at rare and Finnish enriched variants**. To determine which of the 2030 association signals are distinct from previous metabolite GWAS findings, we repeated metabolite association analysis conditioning on all variants that were (a) ≤ 1 Mb of the index variant and (b) previously reported as associated with any metabolite in a curated list of 381 publications ("Methods"; Supplementary Data 6). 303 association signals at 229 index variants remained significant for 201 metabolites ($P_{condition} < 7.2 \times 10^{-11}$; Fig. 3b; Supplementary Data 3). The 303 novel signals included 64 signals (for 58 metabolites) at 51 rare variants and 79 signals

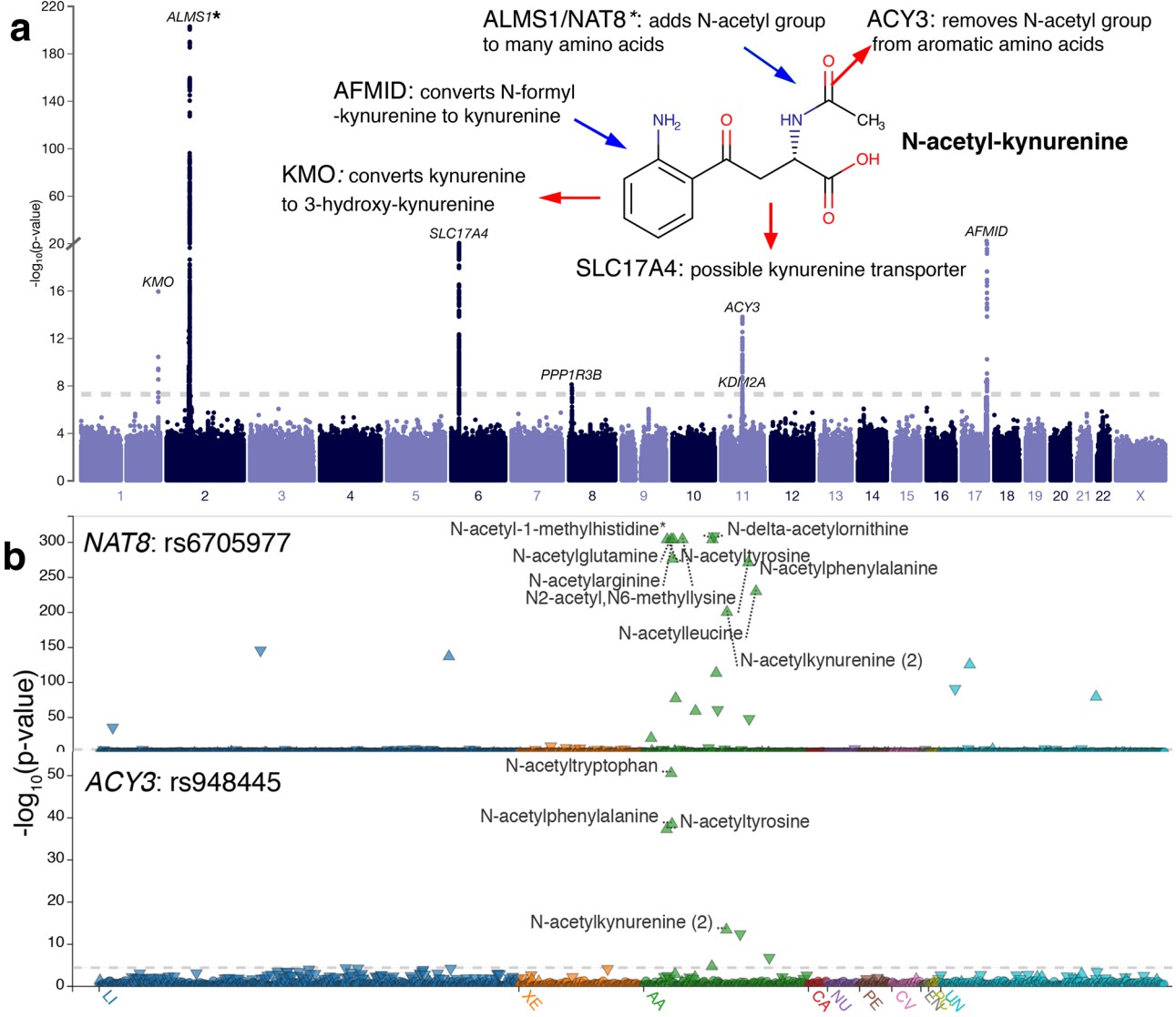

**Fig. 2 Our METSIM Metabolomics PheWeb facilitates the characterization of genetic associations and gene activities. a** Manhattan plot for N-acetylkynurenine highlights the roles of the associated genes (https://pheweb.org/metsim-metab/pheno/C100006378). Chemical structure for N-acetylkynurenine (in bold face) and activities for the associated genes are added manually on top of the Manhattan plot. **b** Stacked PheWeb plots show significant associations between rs6705977 (*NAT8*, https://pheweb.org/metsim-metab/variant/2:73622043-C-G) and fifteen N-acetylated molecules, and the more restricted set of associations between rs948445 (*ACY3*, https://pheweb.org/metsim-metab/variant/11:67647021-C-T) and four N-acetylated aromatic amino acids. LI: lipid; XE: xenobiotics; AA: amino acid; CA: carbohydrate; NU: nucleotide; PE: peptide; CV: cofactor and vitamin; EN: energy; PC: partially characterized; UN: unnamed.

(for 71 metabolites) at 47 variants > 10-fold more frequent in METSIM than in NFE (Table 2); 33 of these signals are at variants both rare and enriched in Finns (Supplementary Data 3). In addition, 17 signals for 16 metabolites are on the X chromosome, and 78 signals are for 44 metabolites identified since 2015 on the Metabolon DiscoveryHD4 platform.

Multiple novel associations arose at the same index variants. For example, we identified novel association signals with 19 metabolites at the putatively-deleterious*SLC23A3* missense variant p.Asn336Lys (rs192756070; Supplementary Fig. 7). p.Asn336Lys has 107-fold greater frequency in METSIM than in NFE (MAF = 2.3% vs. 0.022%) and is likely the causal variant for most or all 19 metabolite associations (VPIP median = 0.98, range = 0.57–1.00). *SLC23A3* encodes an SLC23 ascorbic acid transporter without demonstrated nucleobase transport[22]. These novel associations suggest a wide range of transport functions for SLC23A3.

Among the novel association signals at index variants enriched in Finns, we identified an association with 3-amino-2-piperidone at the putatively-deleterious*OAT* missense variant p.Leu402Pro (rs121965043, $\beta = 1.91$, $P = 3.7 \times 10^{-35}$). p.Leu402Pro has 100-fold greater frequency in METSIM than in NFE (MAF = 0.35% vs. 0.0031%) and is the likely causal variant for this association (VPIP = 0.997). *OAT* encodes the key mitochondrial enzyme ornithine aminotransferase which converts arginine and ornithine into glutamate and gamma aminobutyric acid[23]. *OAT* has not previously been implicated in metabolite GWAS, but inactivation of OAT is responsible for the Finnish heritage disease gyrate atrophy characterized by hyperornithinemia[24]. Previous studies have found increased 3-amino-2-piperidone levels in the urine of individuals with gyrate atrophy[25].

Among the novel association signals on the X chromosome, we identified an association for tiglylcarnitine at the putatively-deleterious*HSD17B10* missense variant p.Ala95Thr (rs201378370,

**Table 1 Summary of the 2030 genetic association signals by metabolite biochemical class.**

| Biochemical class and abbreviation | Total metabolites | Significant metabolites | Total signals | Novel signals |
|---|---|---|---|---|
| Lipid (LI) | 548 | 357 | 903 | 74 |
| Amino acid (AA) | 215 | 154 | 441 | 73 |
| Xenobiotics (XE) | 163 | 52 | 91 | 16 |
| Nucleotide (NU) | 42 | 26 | 65 | 29 |
| Peptide (PE) | 42 | 18 | 28 | 7 |
| Cofactors and vitamins (CV) | 38 | 25 | 69 | 27 |
| Carbohydrate (CA) | 25 | 20 | 38 | 7 |
| Energy (EN) | 10 | 4 | 11 | 3 |
| Partially characterized (PC) | 16 | 8 | 20 | 1 |
| Unnamed (UN) | 292 | 139 | 364 | 66 |
| Total | 1,391 | 803 | 2,030 | 303 |

Significant metabolites: number of metabolites with at least one association signal at $P < 7.2 \times 10^{-11}$.

$\beta = 0.94$, $P = 5.2 \times 10^{-122}$). p.Ala95Thr has 76-fold greater frequency in METSIM than in NFE (MAF = 2.6% vs. 0.034%) and is the likely causal variant for this association (VPIP = 0.997). *HSD17B10* encodes 17-β-hydroxysteroid dehydrogenase X, a mitochondrial enzyme which catalyzes oxidation of neuroactive steroids and degradation of isoleucine[26]. Mutations in *HSD17B10* that abolish enzyme activity lead to HSD10 deficiency, an infantile neurodegenerative disorder in which tiglylcarnitine level is elevated[27]. In contrast, *HSD17B10* is overexpressed in brains of individuals with Alzheimer's disease[28], in which tiglylcarnitine level is decreased[29].

A previous study reported an association between the *GNPTAB* intronic variant rs7964859 and aspartate[15]. We identified an independent aspartate association signal with the *GNPTAB* frameshift variant p.Cys528ValfsTer19 (rs1209353188) (LD $r^2 = 0.01$; $\beta = 0.91$, $P_{condition} = 5.2 \times 10^{-15}$ conditioning on rs7964859). p.Cys528ValfsTer19 is rare in METSIM (MAF = 0.58%), absent in gnomAD NFE, and is the likely causal variant for the METSIM aspartate association (VPIP = 0.996). *GNPTAB* encodes the alpha- and beta-subunits of N-acetylglucosamine-1-phosphotransferase which catalyzes the N-linked glycosylation of asparagine residues with mannose-6-phosphate[30].

**Nominating putative causal genes**. To nominate putative causal genes for metabolite association signals, we applied two approaches. First, we nominated 66 putative causal genes for the 208 association signals for which fine-mapping analysis identified PTV or missense variants at VPIP ≥ 0.8 (see "Fine mapping"). Second, we implemented a knowledge-based approach to integrate biological information about the metabolite and the 20 protein-coding genes nearest the corresponding index variant for the 1666 of the 2030 association signals with named metabolites ("Methods"). The knowledge-based approach nominated 215 single genes for 1033 association signals with 480 metabolites (Supplementary Fig. 8) and 19 sets of 2–7 genes with similar biochemical activity (62 additional genes) for 324 association signals with 208 metabolites (Supplementary Data 3 and 7).

We compared gene nominations for the 138 metabolite association signals for which both approaches nominated causal genes. Compared to the fine-mapping analysis, the knowledge-based approach nominated the same gene for 119 signals, multiple paralogs including the same gene for 18, and a different gene for 1, for an overall consistency > 99% (Supplementary Data 8).

The $277 = 215 + 62$ genes identified by the knowledge-based approach and the 66 identified by fine mapping together

comprised 290 genes. 204 (70%) of the 290 genes are the closest genes to the index variants, including 188 (68%) of the 277 genes identified by the knowledge-based approach. 58 of the 290 genes have not previously been implicated in metabolite GWAS (Fig. 3c). Of the 58 novel genes, 51 were identified by the knowledge-based approach (Supplementary Data 7), 21 by fine-mapping analysis (Supplementary Data 5), and 14 by both. Of the 58 novel genes, 40 represent novel loci and 18 are within loci in which metabolite associations have previously been identified but the genes we nominated have not previously been implicated.

Novel genes nominated based on association signals with amino acid levels provide insight into how the encoded enzymes or transporters contribute to modifications of amino acid derivatives. As a first example, we identified a novel association at the *HDAC6* missense variant p.Arg832His (rs61735967) with N6-acetyllysine (MAF = 2.9%, $\beta = 0.71$, $P = 3.6 \times 10^{-80}$) and suggested p.Arg832His is the likely causal variant (VPIP = 0.998). Both the fine-mapping and knowledge-based approaches nominated *HDAC6* as the putative causal gene. *HDAC6* encodes a lysine deacetylase that removes the acetyl group from acetyllysine in histones. Increased *HDAC6* expression has been found in brains of individuals with Alzheimer's disease[31]. Elevated levels of N6-acetyllysine were recently found in an Alzheimer's disease mouse model[32].

As a second example, we identified a novel association between the *QPCT* intronic variant rs77684493 and pyroglutamylglutamine (MAF = 6.2%, $\beta = -0.55$, $P = 9.0 \times 10^{-31}$). rs77684493 is in near-perfect LD ($r^2 = 0.996$) with the putatively-deleterious *QPCT* missense variant p.Arg54Trp (rs2255991), which was also associated with pyroglutamylglutamine ($\beta = -0.54$, $P = 1.6 \times 10^{-30}$) and has > 7-fold greater frequency in METSIM than in NFE (MAF = 6.3% vs. 0.89%). Our knowledge-based approach nominated *QPCT* as the putative causal gene for this association. *QPCT* encodes the enzyme glutaminyl-peptide cyclotransferase, which performs cyclization of the N-terminal glutamine residues and results in the pyroglutamine residue[33]. *QPCT* has been implicated in a schizophrenia GWAS[34] and suggested as a druggable target for Huntington's disease[35].

GWAS of metabolites recently identified on the Metabolon platform helped nominate novel putative causal genes with high biochemical relevance in known metabolite-associated regions. For example, a previous study in a Japanese sample identified associations for blood creatinine and uracil levels at the *LRIG1* missense variant p.Thr792Met (rs202007714)[36], which is monomorphic in METSIM and gnomAD Finns. We identified an association at the nearby (13 kb) *SLC25A26* missense variant p.Thr208Met (rs13874) with 2,3-dihydroxy-5-methylthio-4-pentenoate (DMTPA) (MAF = 48.3%, $\beta = 0.17$, $P = 2.3 \times 10^{-21}$). We suggest *SLC25A26* as the causal gene for the DMTPA association. DMTPA, an S-adenosylmethionine, was recently identified on the Metabolon DiscoveryHD4 platform. SLC25A26 is the only known mitochondrial S-adenosylmethionine transporter.

Seven of the 58 novel genes were identified only by fine mapping. Among them, we identified a novel association for glycocholenate sulfate at the rare *ADCK5* missense variant p.Ala508Thr (rs552968665; $\beta = 1.31$, $P = 3.6 \times 10^{-12}$), which is > 79-fold more frequent in METSIM than in NFE (MAF = 0.25% vs. 0.0031%). Fine-mapping analysis suggested p.Ala508Thr is the likely causal variant (VPIP = 0.89), implicating a causal role for *ADCK5*. *ADCK5* encodes the aarF domain containing kinase 5. These results suggest *ADCK5* plays a role in human bile acid metabolism.

**Colocalization of metabolites with human diseases**. Integrating metabolite and disease genetic associations can improve fine-mapping resolution[37] and clarify the potentially causal variants and disease genes. We performed Bayesian colocalization

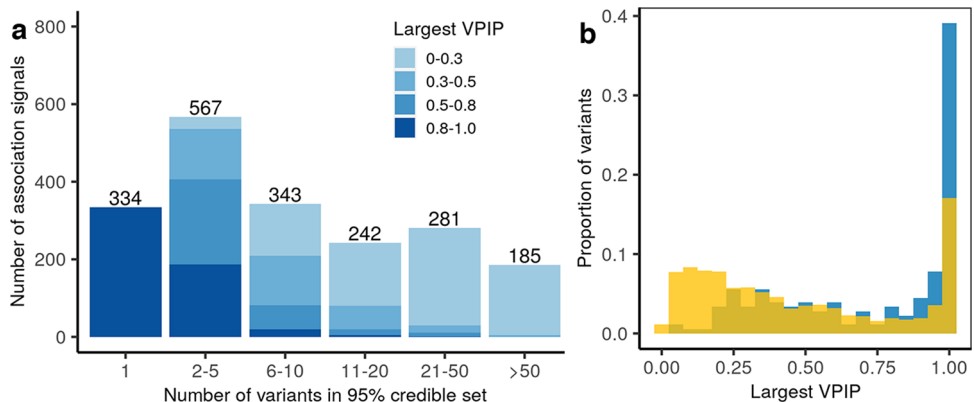

**Fig. 3 Characterization of the 2030 significant metabolite genetic association signals.** Comparison of MAFs for the 1143 index variants between METSIM and non-Finnish Europeans in gnomAD v3.1; index variants are colored **a** purple if MAF>10-fold greater in METSIM than in non-Finnish Europeans; or **b** blue if they represent novel association signals. The dashed line is of slope one through the origin. **c** Overlaid Manhattan plots of the 1391 metabolite GWAS. The red dashed line depicts genome-wide significance threshold $P = 7.2 \times 10^{-11}$. The associations at 40 novel putative causal genes within novel regions (blue) and 18 novel putative causal genes within previously reported regions (maize) are highlighted. The seven novel putative causal genes implicated only by fine-mapping analysis are starred. *HADHA/B* represents the *HADHA* and *HADHB* genes and *ARSD/L* the *ARSD* and *ARSL* genes.

**Fig. 4 The 1952 of the 2030 metabolite genetic association signals identified in stepwise conditional tests with SPIP ≥ 0.95 in DAP-g Bayesian fine mapping. a** Numbers of variants in the 95% credible sets and distribution of variant posterior inclusion probabilities (VPIPs) for the most likely causal variants within the 95% credible sets. **b** Density plot of largest VPIPs highlights the variants with >10-fold greater frequency in METSIM than non-Finnish Europeans (gnomAD v3.1; blue) have larger VPIPs than all other variants (maize).

**Table 2 Top metabolite genetic association signals at 47 novel variants with MAF>10-fold greater in METSIM than in gnomAD v3.1 non-Finnish Europeans.**

| Biochemical name | rsID | MAF(%) | MAF$_{NFE}$(%) | R$_{MAF}$ | β | P | Gene |
|---|---|---|---|---|---|---|---|
| X – 17676 | rs200711248 | 0.321 | 0.009 | 34.5 | 1.18 | 5.40E−11 | – |
| Uracil | rs1254152519 | 0.613 | 0.002 | 395.2 | 1.33 | 5.39E−28 | DPYD |
| palmitoyl dihydrosphingomyelin (d18:0/16:0) | rs752521494 | 0.190 | 0.000 | ∞ | 1.91 | 2.53E−14 | DEGS1 |
| X – 12127 | rs189344406 | 0.302 | 0.006 | 48.7 | −1.75 | 3.27E−14 | – |
| Campesterol | rs1247627279 | 0.362 | 0.003 | 116.8 | 1.11 | 5.26E−11 | ABCG5\|ABCG8 |
| Xanthurenate | rs199546957 | 0.151 | 0.000 | ∞ | 2.07 | 4.96E−18 | KYNU |
| hydantoin-5-propionate | rs144419430 | 0.928 | 0.015 | 59.9 | −0.76 | 2.16E−13 | Unknown |
| Glycerate | rs192756070 | 2.917 | 0.022 | 134.5 | −1.07 | 8.04E−85 | SLC23A3 |
| X – 15666 | rs140758280 | 4.073 | 0.376 | 10.8 | −0.42 | 1.40E−19 | – |
| X – 24475 | rs202158371 | 0.814 | 0.012 | 65.6 | −0.91 | 9.61E−17 | – |
| methyl glucopyranoside (alpha + beta) | rs186284085 | 1.370 | 0.008 | 176.7 | 0.69 | 9.41E−20 | GBA3 |
| X – 12844 | rs200280202 | 1.436 | 0.034 | 42.1 | −0.83 | 5.21E−26 | – |
| N-acetylglucosaminylasparagine | rs561604250 | 0.668 | 0.008 | 86.3 | 0.97 | 1.16E−14 | AGA |
| X – 24544 | rs141884785 | 1.320 | 0.073 | 18.1 | 0.76 | 1.74E−22 | – |
| Choline | rs200164783 | 2.634 | 0.136 | 19.3 | 0.70 | 2.32E−33 | Unknown |
| Serine | rs1297328831 | 0.162 | 0.002 | 104.6 | −1.64 | 6.82E−13 | Unknown |
| N-acetylhistidine | rs146438324 | 0.946 | 0.057 | 16.5 | 1.51 | 1.71E−61 | NAT16 |
| Sulfate | rs138989506 | 1.910 | 0.029 | 64.9 | −1.41 | 1.55E−99 | SLC13A1 |
| X – 26054 | rs976212663 | 0.231 | 0.019 | 12.4 | −2.08 | 3.52E−15 | – |
| 3-(3-amino-3-carboxypropyl)uridine | rs149926554 | 2.898 | 0.175 | 16.6 | 0.43 | 7.97E−14 | Unknown |
| 5-oxoproline | rs782359519 | 0.854 | 0.019 | 45.8 | 0.92 | 1.43E−24 | OPLAH |
| alpha-ketoglutarate | rs191616586 | 3.113 | 0.195 | 15.9 | −0.45 | 7.65E−17 | Unknown |
| 5-oxoproline | rs558946866 | 0.442 | 0.034 | 13.0 | 0.99 | 4.54E−15 | OPLAH |
| 3beta-hydroxy-5-cholestenoate | rs552968665 | 0.245 | 0.003 | 79.1 | 1.55 | 8.62E−16 | Unknown |
| 3-amino-2-piperidone | rs121965043 | 0.346 | 0.003 | 111.7 | 1.91 | 3.73E−35 | OAT |
| Deoxycarnitine | rs1268699195 | 0.577 | 0.002 | 372.3 | 0.90 | 1.25E−12 | Unknown |
| beta-citrylglutamate | rs182295429 | 4.615 | 0.064 | 72.6 | 0.52 | 3.73E−28 | NAALAD2 |
| Betaine | rs1358634021 | 0.093 | 0.005 | 20.0 | 2.43 | 1.56E−16 | SLC6A12 |
| Aspartate | rs1209353188 | 0.578 | 0.000 | ∞ | 0.91 | 9.92E−14 | GNPTAB |
| Succinylcarnitine (C4-DC) | rs200127857 | 0.342 | 0.005 | 73.6 | 2.20 | 2.37E−45 | LACTB |
| Succinylcarnitine (C4-DC) | rs200480788 | 0.104 | 0.005 | 22.4 | 2.40 | 3.96E−19 | LACTB |
| 5-hydroxylysine | rs201135688 | 4.513 | 0.447 | 10.1 | 0.39 | 6.83E−22 | HYKK |
| X – 17676 | rs185603444 | 1.983 | 0.184 | 10.8 | −0.60 | 5.56E−16 | – |
| Orotidine | rs201899452 | 0.143 | 0.002 | 92.3 | −1.80 | 2.41E−13 | NT5C |
| N-formylanthranilic acid | rs77585764 | 5.407 | 0.378 | 14.3 | 1.24 | 9.55E−218 | AFMID |
| Sphingomyelin (d18:1/18:1, d18:2/18:0) | rs527480139 | 0.330 | 0.000 | ∞ | −1.41 | 2.51E−19 | CERS4 |
| Glycosyl-N-stearoyl-sphingosine (d18:1/18:0) | rs1013893365 | 2.956 | 0.046 | 64.2 | 1.25 | 2.17E−106 | CERS1 |
| X – 11315 | rs201742362 | 1.134 | 0.033 | 34.9 | 1.43 | 1.62E−61 | – |
| 2-O-methylascorbic acid | rs6267 | 5.778 | 0.142 | 40.6 | −0.82 | 9.49E−108 | COMT |
| 2-O-methylascorbic acid | rs199637204 | 0.098 | 0.005 | 21.1 | −1.87 | 1.49E−12 | COMT |
| Gamma-tocopherol/beta-tocopherol | rs182488695 | 1.548 | 0.008 | 199.9 | 0.89 | 7.37E−33 | SEC14L2 |
| N6-succinyladenosine | rs8192461 | 1.175 | 0.073 | 16.1 | 1.12 | 9.38E−27 | ADSL |
| N6-succinyladenosine | rs773404017 | 0.414 | 0.006 | 66.8 | 2.55 | 2.30E−44 | ADSL |
| 5-methyluridine (ribothymidine) | rs548223694 | 0.448 | 0.039 | 11.6 | 1.36 | 9.47E−27 | TYMP |
| 5-methyluridine (ribothymidine) | rs756647111 | 0.119 | 0.000 | ∞ | 2.04 | 2.35E−14 | TYMP |
| 2'-deoxyuridine | rs556167510 | 0.601 | 0.042 | 14.4 | 1.07 | 2.45E−14 | TYMP |
| Tiglylcarnitine (C5:1-DC) | rs201378370 | 2.584 | 0.034 | 76.7 | 0.94 | 5.21E−122 | HSD17B10 |

Biochemical name: biochemical name of the metabolite. rsID: dbSNP variant ID. MAF, MAF$_{NFE}$, and R$_{MAF}$: minor allele frequency in METSIM, in gnomAD v3.1 non-Finnish Europeans, and their ratio. When the index variant is monomorphic in gnomAD v3.1 non-Finnish Europeans (n = 34,029), the ratio is labeled as infinite, ∞. β: effect size estimate from the metabolite-specific stepwise conditional association test. P: p-value of metabolite-specific stepwise conditional association test. Gene: the putative causal gene(s) nominated in the knowledge-based approach. Gene symbols are italic. For unnamed metabolites, the putative causal gene results are represented by "–". If no putative causal gene is nominated, it is labeled as "unknown". If multiple putative causal genes are nominated, they are separated by a vertical bar.

analysis[37,38] based on the probabilistic fine-mapping results of METSIM metabolites and of 980 disease and disease-related dichotomous traits (henceforth, disease traits) in 176,899 Finns in FinnGen release 4 ("Methods"; Supplementary Data 9). We calculated the regional colocalization probability (RCP) of a putative causal variant shared between a METSIM metabolite and a FinnGen disease trait ("Methods"). We identified 946 colocalizations involving 248 metabolites and 105 interrelated disease traits (RCP ≥ 0.5; Supplementary Data 10).

Integrating metabolite associations substantially increased the fine-mapping confidence of FinnGen disease trait association signals (SPIP median = 0.91 vs. 0.73; paired t-test $P = 2.5 \times 10^{-70}$) and the probability assigned to the most likely disease variant (maximum VPIP median = 0.54 vs. 0.11; paired t-test $P = 9.1 \times 10^{-128}$; Supplementary Fig. 9). For example, the putatively-deleterious SERPINA1 missense variant p.Glu366Lys (rs28929474) was associated with N-acetylglucosaminylasparagine level in METSIM (MAF = 2.3%, β = 0.56, $P = 4.9 \times 10^{-16}$) and risk of cholestasis of pregnancy

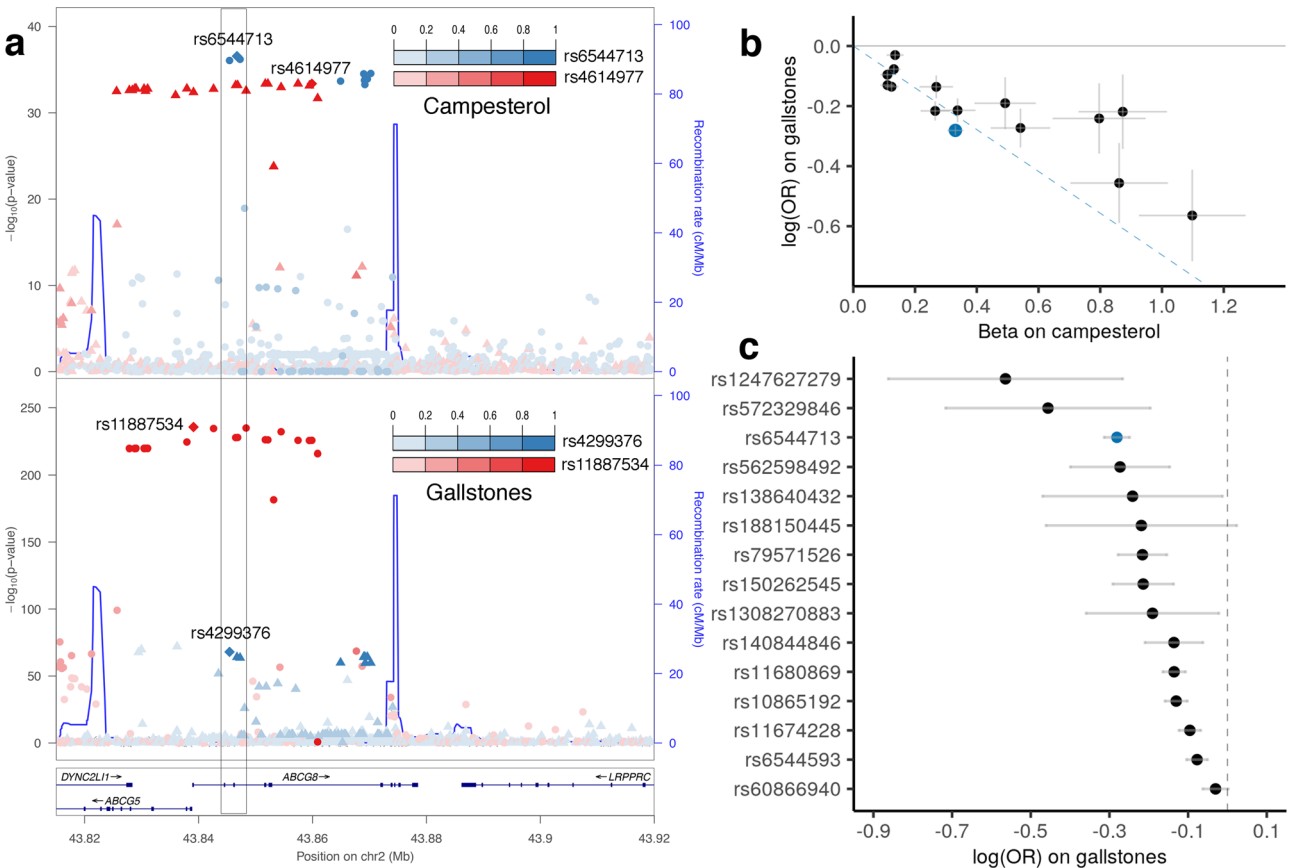

**Fig. 5 Colocalization and causal relationship between campesterol and gallstones. a** Stacked regional association plots for campesterol and gallstones (cholelithiasis, K11_CHOLELITH in FinnGen release 4) in the *ABCG5/ABCG8* region. The index variants identified in stepwise conditional analysis (campesterol) and approximate conditional analysis (gallstones) are labeled and variants colored by their linkage disequilibrium (LD) to the index variant with which they are in strongest LD in METSIM. The campesterol signal (index variant rs6544713) is colocalized with the gallstone signal (rs4299376, pairwise LD $r^2 = 0.993$, RCP = 0.65) shown in the gray box. In contrast, no colocalization was detected between the signals indexed by rs4614977 and rs11887534. No coding variants within 1 Mb have LD $r^2 > 0.2$ with rs6544713 in METSIM. **b** Comparison of effect sizes for the 15 instrumental variables genome-wide without significant heterogeneity ($P > 0.05$) used in Mendelian randomization analysis between campesterol and gallstones. rs6544713 is in blue. The slope of the blue dashed line depicts the estimated causal effect size of campesterol on gallstones. The Egger regression intercept is deemed not significant ($P = 0.15$). **c** Negative relationship between instrumental variable and risk of gallstones. OR: odds ratio.

in FinnGen (odds ratio (OR) = 6.23, $P = 8.1 \times 10^{-17}$). Our fine-mapping analysis suggested a causal role of *SERPINA1* p.Glu366Lys for N-acetylglucosaminylasparagine (VPIP = 0.81). We detected colocalization in this region between signals for N-acetylglucosaminylasparagine and cholestasis of pregnancy (RCP = 0.99). Colocalizing these association signals increased the SPIP for cholestasis of pregnancy from 0.69 to 0.99, and the VPIP of *SERPINA1* p.Glu366Lys from 0.37 to 0.80. *SERPINA1* encodes a serine protease inhibitor produced mainly in the liver. *SERPINA1* mutations have been associated with familial intrahepatic cholestasis[39], and *SERPINA1* p.Glu366Lys with liver diseases[40] and circulating liver enzymes[41]. We may have missed colocalizations where the true causal variants were discarded in METSIM or FinnGen; in such cases, we may have detected colocalizations at other, likely more common, variants.

**Campesterol and gallstones: potential causal link**. Gallstones affect 10–20% of adults worldwide[42]. Aberrant cholesterol homeostasis, particularly the physical–chemical imbalance of cholesterol solubility in bile, induces gallstones[43]. Blood campesterol levels have been associated with gallstones[44], but it is uncertain whether the relationship is causal. We identified associations at the *ABCG8* intronic variant rs6544713 with lower

campesterol level in METSIM (MAF = 20.1%, $\beta = -0.33$, $P = 2.7 \times 10^{-37}$) and higher gallstone risk in FinnGen (OR = 1.32, $P = 8.0 \times 10^{-65}$). In 4689 METSIM participants with observed campesterol levels, 199 with gallstones, plasma campesterol level was inversely associated with gallstone risk ($\beta = -0.52$, $P = 3.7 \times 10^{-5}$).

Colocalization analysis suggested campesterol and gallstones share the same causal variant in this region (RCP = 0.65; Fig. 5a) and nominated rs6544713 (SCP = 0.45) as the most likely causal variant. rs6544713 resides in active regulatory units in intestinal tissue[45] and its campesterol-decreasing allele is associated with higher *ABCG8* expression in colon tissue[46] ($\beta = 0.37$, $P = 7.2 \times 10^{-16}$), but not in the liver[46]. *ABCG8* and its nearby paralog *ABCG5* have previously been suggested as candidate genes for gallstones[47]. Mutations in *ABCG5* and *ABCG8* cause sitosterolemia, characterized by elevated campesterol[48].

Mendelian randomization analysis using 15 independent variants for campesterol suggested a causal effect of lower plasma campesterol level on higher gallstone risk ("Methods"; $\beta = -0.70$, $P = 7.2 \times 10^{-8}$; Fig. 5b, c). *ABCG5* and *ABCG8* together encode a heterodimeric ATP-binding cassette transporter that facilitates secretion of cholesterol and non-cholesterol sterols in the intestine and bile. High plasma campesterol levels might compete with cholesterol for ABCG5/ABCG8 transporters during biliary

cholesterol secretion, resulting in decreased biliary cholesterol levels and reduced risk of gallstones[49,50] (Supplementary Fig. 10).

**DBH influence on vanillylmandelate and hypertension: distinct pathways**. Stepwise conditional analysis identified the putatively-deleterious *DBH* missense variant p.Arg79Trp (rs77273740) as associated with lower vanillylmandelate ($\beta = -0.38$, $P = 1.8 \times 10^{-15}$); p.Arg79Trp is > 10-fold more frequent in METSIM than in NFE (MAF = 4.5% vs. 0.39%). Both fine-mapping and the knowledge-based approach suggested a causal role for *DBH*. The knowledge-based approach suggested DBH could exhibit an effect on vanillylmandelate in two ways (Supplementary Fig. 10): by converting dopamine to norepinephrine, a vanillylmandelate precursor[51], or by transforming homovanillate acid to vanillylmandelate through hydroxylation[52].

In FinnGen, the *DBH* p.Arg79Trp vanillylmandelate-decreasing allele was significantly associated with lower hypertension risk (OR = 0.84, $P = 5.2 \times 10^{-13}$), consistent with previous associations with systolic and diastolic blood pressures[53,54]. No other variants were associated with FinnGen hypertension or METSIM vanillylmandelate in this region ($P > 10^{-6}$; Supplementary Fig. 11). In 5173 METSIM participants with observed vanillylmandelate levels, 1073 with hypertension, plasma vanillylmandelate level was weakly but significantly associated with hypertension ($\beta = 0.52$, $R^2 = 0.01$, $P = 3.2 \times 10^{-9}$). Controlling for hypertension status gave nearly identical genetic association between *DBH* p.Arg79Trp and vanillylmandelate ($\beta = -0.38$, $P = 1.6 \times 10^{-15}$).

Colocalization analysis suggested that hypertension colocalized with vanillylmandelate (RCP = 0.996). Using Mendelian randomization, we did not find significant evidence of causal effects of vanilymandelate on hypertension ($P = 0.15$; 10 independent variants; Supplementary Fig. 11) or of hypertension on vanillylmandelate ($P = 0.17$; 157 independent variants), suggesting this signal conferred effects on hypertension risk and vanillylmandelate through different pathways, consistent with the two possible DBH effects identified by our knowledge-based approach (Supplementary Fig. 10). The analysis from vanillylmandelate to hypertension could only make use of ten instruments and so may be underpowered.

## Discussion

We performed GWAS of 1391 plasma metabolites in 6136 men from the late-settlement region of Finland. We sought to identify putative causal variants and genes for the resulting genetic associations, and interrogated disease molecular mechanisms by integrating metabolite and disease genetic associations. We identified 2030 association signals for 803 metabolites, including 157 signals for 125 metabolites at 121 rare variants. We identified 303 association signals for 201 metabolites as novel, including 64 signals for 58 metabolites at 51 rare variants.

Over half of these 303 novel association signals stem from the population history of Finland, the analysis of previously-unstudied metabolites, or the analysis of the X chromosome. The Finnish population history of alternating founding events and population expansions has resulted in a set of genetic variants rare elsewhere but more common in Finns, providing increased statistical power for genetic discovery for these variants[2], as exemplified by the Finnish heritage diseases[55]. 79 of the 303 novel association signals we identified are at 47 variants with MAF > 10-fold greater in METSIM than in NFE, with 37 novel signals at 14 variants with MAF > 100-fold greater. These include the novel association of 3-amino-2-piperidone with the rare *OAT* missense variant p.Leu402Pro; mutations in *OAT* cause the Finnish heritage disease gyrate atrophy (see "Results").

Metabolon continues to expand the set of metabolites identified on their platform. 78 of the 303 novel association signals were for 44 metabolites identified after 2015 on the Metabolon DiscoveryHD4 platform, and so studied only in the most recent Metabolon-based metabolomics GWAS[13]. For example, we identified a novel association at *SLC23A3* missense variant p.Asn336Lys for 2-O-methylascorbic acid, identified on the Metabolon platform in 2019.

Our study is one of the first Metabolon metabolomics GWAS to analyze the X chromosome, where 17 of the 303 novel association signals arose. For example, we identified a novel association for tiglylcarnitine at the *HSD17B10* missense variant p.Ala95Thr. *HSD17B10* mutations cause a rare inborn error of metabolism characterized by cognitive impairment and variable neurological abnormalities.

Biochemical analysis existed for decades prior to the advent of GWAS. Experiments linking a gene to a metabolite often already existed in the published literature. We identified 277 putative causal genes through existing links in the literature between tested metabolites and biochemical activities of genes near our association signals. Our results suggested most of these putative causal genes acted on the associated metabolites or closely-related metabolites. These putative causal genes characterized the genetic regulatory mechanisms for plasma metabolite levels. The associations of multiple metabolites with the same gene help improve the understanding of the gene function. For example, we nominated *SLC23A3* as a causal gene for 19 metabolites of various biochemical classes, suggesting a wide range of transport functions in addition to its known role as an ascorbic acid transporter.

Integrating metabolite and disease genetic associations helps disentangle disease biology. We identified 946 metabolite-disease trait pairs likely sharing the same causal variants, which helped pinpoint the likely causal variants and disease genes (Supplementary Data 10). For example, colocalization analysis of acetylglucosaminylasparagine and cholestasis suggested a shared causal role of *SERPINA1* p.Glu366Lys. Mendelian randomization analysis suggested for the first time a protective effect of high plasma campesterol on gallstones. Plasma campesterol is commonly used as a biomarker for gallstones[56] and campesterol is used as a supplement to reduce low-density lipoprotein cholesterol[57]. Our finding provides supporting evidence for these applications of campesterol in the treatment of gallstones.

Data sharing increases the impact of genetic studies. To support data exploration of our metabolite GWAS results[58], we have constructed a METSIM metabolite PheWeb site[59] (Fig. 2). This site supports querying, visualizing, and downloading our METSIM Metabolon metabolite genetic association results, including Manhattan and quantile-quantile plots, and summary statistics for all 1391 metabolites. In addition, we provide direct links to the Human Metabolome Database (HMDB)[60], which presents the metabolites' biochemical characteristics and enables interpretation of metabolite genetic association results.

In summary, we performed parallel GWAS for 1391 plasma metabolites in 6136 adult Finnish males from the METSIM study, colocalized metabolite and disease genetic associations, and made these GWAS results available using PheWeb. Our findings reveal genetic determinants for a wide range of plasma metabolites and demonstrate the utility of metabolite genetic associations for the investigation of disease biology.

## Methods

**metabolic syndrome in men (METSIM) study**. METSIM is a study of 10,197 Finnish men from Kuopio in the late-settlement region of northeast Finland designed to investigate factors associated with type 2 diabetes and cardiovascular diseases[3] (Supplementary Table 2). Participants were aged 45–74 (median = 58) years during baseline visits from 2005 to 2010. Participants provided demographic, diet, exercise, disease, and medication history information, and underwent laboratory measurements, including oral glucose tolerance test, after ≥10-hour overnight fast. Morbidity, mortality, and drug treatment information was

periodically updated for participants who consented to use of their hospital admission, drug reimbursement, and prescription records in Finnish national registries. Due to funding constraints, we randomly selected 6490 of the 8777 METSIM participants who at baseline were neither diagnosed with diabetes nor taking diabetes medications that might broadly impact metabolomics levels for the Metabolon metabolomics assay. After exclusion of participants who subsequently developed diabetes ($n = 264$), lacked array genotypes ($n = 65$) or body mass index (BMI) measurement ($n = 1$), had sex mismatch ($n = 3$), and/or were non-Finnish ($n = 21$), our analysis set comprised 6136 participants (Supplementary Table 2; Fig. 1). This study was approved by the Ethics Committee at the University of Eastern Finland and the Institutional Review Board at the University of Michigan. All participants provided written informed consent.

**Metabolomics profiling and data processing.** Non-targeted metabolomics profiling was performed at Metabolon, Inc. (Durham, North Carolina, USA)[61] on EDTA-plasma samples obtained after ≥10-h overnight fast during METSIM baseline visits. Briefly, methanol extraction of biochemicals followed by a non-targeted relative quantitative liquid chromatography–tandem mass spectrometry (LC-MS/MS) Metabolon DiscoveryHD4 platform was applied to assay named ($n = 1240$) and unnamed ($n = 304$) metabolites (Supplementary Table 1 and Supplementary Data 1). Samples were randomized across batches. Batches contained ~144 METSIM samples and 20 well-characterized human-EDTA plasma samples for quality control (QC). All 6490 samples were processed together for peak quantification and data scaling. We quantified raw mass spectrometry peaks for each metabolite using the area under the curve. We evaluated overall process variability by the median relative standard deviation for endogenous metabolites present in all 20 technical replicates in each batch. We adjusted for variation caused by day-to-day instrument tuning differences and columns used for biochemical extraction by scaling the raw peak quantification to the median for each metabolite by Metabolon batch.

**Array genotyping and exome sequencing.** All METSIM participants were array genotyped on the Human OmniExpress-12v1_C BeadChip (OmniExpress) and Infinium HumanExome-12 v1.0 BeadChip (exome array) platforms[62]. We excluded individuals for sex or relationship mismatch, apparent sample duplication, or ancestry outliers based on genetic principal component analysis (PCA). We removed variants with genotype call rate < 95% (OmniExpress) or < 98% (exome array), or Hardy-Weinberg equilibrium (HWE) $P < 10^{-6}$ (either array)[62].

We captured exomes for all METSIM participants by SeqCap EZ HGSC VCRome kit (Roche) and sequenced them by HiSeq2000 (Illumina) (average depth 45×)[2]. For exome sequences, we excluded samples with estimated contamination > 3% or sample swaps compared to the array genotype data[62] and required single-nucleotide variant (SNV) array genotype concordance > 90% if array data were available. We filtered variants with genotype call rate < 98%, HWE $P < 10^{-6}$, or overall low allele balance (alternate allele count/sum of total allele count < 30%)[2]. The resulting array-genotype dataset consisted of n = 10,066 METSIM participants with 679,866 SNVs. The exome-sequence dataset consisted of $n = 9957$ participants with 583,947 SNVs and 40,270 small insertions/deletions (indels).

**Genome sequencing.** We whole genome sequenced METSIM participants in two waves. In wave 1, we genome sequenced 3074 METSIM participants (average depth 23x)[63]. Genomic DNA was fragmented on a Covaris LE220 instrument and size-selected to ensure an average insert size of 350–375 base pairs (bp). Libraries were constructed with the Illumina TruSeq or KAPA Hyper PCR-free library prep kit. qPCR was used to determine concentration of each library. Libraries were subsequently pooled and sequenced with 2 × 150 bp paired-end reads using HiSeq X (Illumina). We filtered read alignments with mismatch rate ≥5%, inter-chromosomal rate ≥5%, discordance rate of paired reads ≥5%, or haploid coverage < 19.5x. We generated QC statistics in Picard v2.4.1 (http://broadinstitute.github.io/picard/), Samtools v1.3.1 (https://github.com/samtools/)[64], and VerifyBamID v1.1.3 (https://github.com/Griffan/VerifyBamID)[65]. We called SNVs and small indels and performed base quality score recalibration in GATK v3.5 (https://gatk.broadinstitute.org/). We excluded variants with missingness > 2%, HWE $P < 10^{-6}$ in unrelated individuals, or allele imbalance < 30%. The resulting genome sequence consisted of $n = 3074$ participants genotyped for 23,849,428 SNVs and 2,914,167 indels. We used wave 1 as part of our imputation reference panel (see "METSIM integrative panel and genotype imputation").

In wave 2, we sequenced 2875 additional METSIM participants using the same methods used for wave 1. We generated a combined wave $1 + 2$ call set of $n = 5949$ using the same methods, resulting in calls for 55,648,111 SNVs and 12,850,837 indels. Wave 2 data became available only after the main analysis for this paper was complete; we used wave $1 + 2$ combined data to determine linkage disequilibrium (LD) proxies for previously-identified metabolite associated variants that were missing in wave 1 but present in wave $1 + 2$ combined data (see "Identification of novel associations").

**METSIM integrative panel and genotype imputation.** Using the 3074 METSIM participants with wave 1 genome sequence data, we generated an integrated list of genetic variant sites by merging site lists from the genome and exome sequence data, and the OmniExpress and exome array data. Of the 3074 participants, 3055 had OmniExpress and exome array data, and 3037 had exome sequence data. We calculated genotype likelihoods for each individual at each site as the product of genotype likelihoods assuming independent data across platforms[66]. For OmniExpress and exome array genotypes, we converted genotype calls to genotype likelihoods assuming a genotype error rate of $10^{-6}$. We then phased genotypes using integrated genotype likelihoods in Beagle v4.1 (https://faculty.washington.edu/browning/beagle/b4_1.html) with 50,000 markers per chunk and 3000 overlapping genetic markers between consecutive chunks[67]. We subsequently excluded 1 individual who self-identified as non-Finnish, 2 individuals identified as population outliers in genetic PCA, and 149 close relatives (estimated kinship ≥ 0.125 in KING v2.2.1 (https://www.kingrelatedness.com)[68]). The resulting integrative panel comprised 2922 individuals genotyped for 23,294,337 SNVs and 2,851,848 indels (Supplementary Table 3). 2670 (91.4%) of the 2922 individuals had Metabolon metabolomics data.

We imputed genotypes for the 6490 study participants on the framework of their OmniExpress genotypes using the METSIM integrative panel with Minimac v4[69]. We excluded imputed variants with imputation $r^2 < 0.3$, leaving 19,182,997 SNVs and 2,404,717 indels for downstream analysis (Supplementary Table 4).

**Variant functional annotation.** We annotated all variants using the Ensembl Variant Effect Predictor (VEP, https://useast.ensembl.org/info/docs/tools/vep/index.html) version 99[70]. We used the "-pick_order" option to annotate each variant using a single transcript, with transcripts prioritized in the following order: transcript support level (i.e., well-supported and poorly-supported transcript models based on the type and quality of the alignments used to annotate the transcript), transcript biotype (protein coding preferred), APPRIS isoform annotation (i.e. annotation based on a range of computational methods to identify the most functionally important transcripts from cross-species conservation), deleteriousness of annotation as estimated by Ensembl, transcript CCDS status (i.e., amount and type of evidence that supports the existence of a variant), canonical status of transcript (https://m.ensembl.org/Help/Glossary), and transcript length[71]. We used the dbNSFP (version 4.0)[72] plugin to generate additional predictions of variant deleteriousness from five in silico algorithms: Polyphen2 HDIV[73], Polyphen2 HVAR[73], SIFT4G[74], MutationTaster[75], and the Likelihood Ratio Test (LRT)[76].

**Trait transformation.** For each metabolite, we inverse normalized the Metabolon-reported metabolite level, regressed on covariates (age at sampling, Metabolon batch, and lipid-lowering medication use status for lipid traits only), and inverse normalized the residuals. In the single-variant association analyses with BMI adjustment, we also included BMI among the covariates.

**Single-variant association analysis.** To account for sample relatedness and potential population stratification among the 6136 participants in our analysis set, we carried out single-variant association tests using a linear mixed model in EPACTS (v3.2.6) (https://github.com/statgen/EPACTS) on the normalized residual metabolite values. We limited analysis to the 1391 metabolites successfully measured on ≥ 500 METSIM participants and to the genetic variants with minor allele count (MAC) ≥ 5; we did not impute missing metabolite data. This resulted in 10,914,098 to 16,172,108 variants (median = 16,042,879) tested across the 1391 metabolites, since the number of variants with MAC ≥ 5 varied with the set of individuals successfully measured for each metabolite.

To choose a study-wise significance threshold for the 1391 parallel metabolite GWAS, we carried out PCA across the metabolites to determine the number of principal components required to explain metabolite variation. To account for missing data (Supplementary Fig. 12), we first imputed missing metabolite values using the K-nearest neighbors approach[77] with K = 5. PCA of the imputed data showed that 692 principal components explained 95% of phenotypic variation for the 1391 metabolites. We therefore used a study-wise significance threshold of $P < 5.0 \times 10^{-8}/692 = 7.2 \times 10^{-11}$ for our single-variant analyses. A metabolite quantified in $n = 500$ participants provided > 80% power at $P < 7.2 \times 10^{-11}$ to detect variants that explained phenotypic variance ≥ 11% and had MAC ≥ 5.

**PheWeb browser.** We built a PheWeb browser[59] of the 1391 metabolite GWAS to support interactive visualization, exploration, and download of these results. This PheWeb (https://pheweb.org/metsim-metab) includes Manhattan and quantile-quantile plots, summary statistics, and links to biochemical characteristics and functions in the Human Metabolome Database (HMDB)[60] for all 1391 metabolites.

**Stepwise conditional analysis.** We carried out stepwise conditional analysis in EPACTS (v3.2.6) (https://github.com/statgen/EPACTS) to identify near-independent association signals. For each metabolite-chromosome pair with at least one single-trait genome-wide significant association ($P < 5.0 \times 10^{-8}$), we first conditioned on the most significant associated variant and continued conditioning on the most significant remaining variant until no variant attained $P < 5.0 \times 10^{-8}$.

**Fine mapping and credible sets**. For each of the 2030 nearly-independent association signals, we built genomic regions of 1 Mb on either side of the index variant, less near chromosome ends. We merged overlapping regions for the same metabolite, resulting in 1501 genomic regions of 1.2 to 3.1 Mb. To identify potential causal variants within each region, we performed fine-mapping analysis using the Deterministic Approximation of Posteriors (DAP-g) method[20] (https://github.com/xqwen/dap), assigning equal priors to all candidate variants. DAP-g uses individual-level metabolite, genotype, and covariate data to produce fine-mapping results. Since DAP-g does not allow for related participants, we corrected for relatedness approximately by including the first ten genetic principal components as covariates; repeating the DAP-g analysis with 0, 20, or 100 principal components yielded similar results.

DAP-g allows for multiple independent association signals within each region. For each identified signal, DAP-g computes (1) a signal posterior inclusion probability (SPIP) that there is at least one causal variant in the signal; and (2) a posterior inclusion probability for each variant (VPIP) that the variant is causal for the signal. For each of the 1952 signals identified in stepwise conditional tests that had SPIP $\geq 0.95$ in DAP-g, we constructed a 95% credible set of potential causal variants by ranking the variants in descending VPIP and including variants until their summed VPIP was $\geq 0.95$.

**Identification of novel associations**. To assess which of our metabolite associations were novel, we compiled a list of 381 published metabolite GWAS papers (Supplementary Data 6): 354 from the NHGRI-EBI GWAS catalog[78] (https://www.ebi.ac.uk/gwas/; release date December 1, 2020); and 27 others from the list curated by Kastenmüller et al.[11] (accessed April 1, 2021). From these papers, we identified 8502 variants with metabolite associations at $P < 5.0 \times 10^{-8}$ or at the significance threshold used in the paper, whichever was more stringent (Supplementary Data 6). Among these 8502 published variants, 7807 were present in the METSIM imputed genotype data. For 194 of the published variants not present in the METSIM imputed genotype data, we identified proxies (LD $r^2 \geq 0.8$ and $\leq$ 500 kb) using the wave $1 + 2$ genome sequence dataset of 5949 METSIM participants. The 7807 variants present in the METSIM imputed genotype data and the 194 LD proxies for missing variants together comprised 8000 unique variants. To avoid problems with multicollinearity, we pruned these 8000 variants at METSIM LD $r^2 > 0.99$ and $\leq 1$ Mb, yielding 6501 LD-pruned variants. Then, for each of the 2030 association signals, we repeated the conditional association analysis including the subset of these LD-pruned variants within $\leq 1$ Mb of the corresponding index variant as covariates. We considered as novel signals those index variants with conditional $P < 7.2 \times 10^{-11}$ and location > 500 kb from any of the $8502 - 7807 - 194 = 501$ published variants, which were neither present nor with proxies in the METSIM imputed genotype data. Among the 501 variants, 380 were monomorphic in gnomAD v3.1 Finns ($n = 5316$).

**Knowledge-based approach to gene nomination**. To nominate putative causal genes for the 1666 of 2030 signals associated with named metabolites, we employed a two-stage knowledge-based approach[15]. In stage 1, for each variant, we identified the 20 closest protein-coding genes using the minimum distance from the index variant to the refSeq genes' transcription start or end sites. We employed an algorithm to look for lexical overlaps between the associated metabolite and each of the 20 genes. Specifically, we searched automatically for matching strings using customized scripts between: (1) the HMDB[60] metabolite name and synonyms and Entrez gene names;[79] (2) the metabolite and Entrez gene names listed in HMDB as interacting with the metabolite; (3) the metabolite name and Uniprot protein names[75] and their synonyms; (4) the metabolite and its parent classes as defined in HMDB and the Uniprot protein names and their synonyms; (5) the metabolite name and rare disease names linked to each gene in OMIM (Online Mendelian Inheritance in Man, https://omim.org/, accessed January 1, 2021) after removing the non-specific substrings uria, emia, deficiency, disease, transient, neonatal, hyper, hypo, defect, syndrome, familial, autosomal, dominant, recessive, benign, infantile, hereditary, congenital, early-onset, idiopathic; (6) the metabolite and its parent classes and Gene Ontology (GO) biological process names[80] associated with each gene after removing the non-specific substrings metabolic process, metabolism, catabolic process, response to, positive regulation of, negative regulation of, regulation of (we only considered gene sets of < 500 genes); and (7) Kyoto Encyclopedia of Genes and Genomes (KEGG)[81] maps (https://www.kegg.jp/) containing the metabolite (as defined in HMDB) and KEGG maps containing each gene (as defined in KEGG) omitting the large "metabolic process map". For each of these pairs of terms, we calculated a Pair Distance score ranging from 0 to 1 using the Ruby gem "fuzzy_match" (https://github.com/seamusabshere/fuzzy_match), and considered a score > 0.5 as a match.

In stage 2, we manually reviewed the evidence collected at stage 1. We selected the biologically most plausible causal gene if we identified experimental evidence linking the gene to the metabolite. >1 putative causal genes could be nominated if > 1 gene was suggested in stage 1 and/or 2; this happened most often when a locus contains multiple paralogs with similar biochemical activity. If no clear experimental evidence existed for any of the 20 genes, no causal gene was selected.

**Colocalization of FinnGen disease traits and METSIM metabolites**. To identify shared causal variants between METSIM metabolites and FinnGen disease traits, we carried out Bayesian pairwise colocalization analysis using fastENLOC[37,38]

(https://github.com/xqwen/fastenloc). We downloaded FinnGen release 4 (https://www.finngen.fi/en/access_results) FINEMAP[82]-based fine-mapping results for 980 disease traits with at least one association at $P < 5.0 \times 10^{-8}$. fastENLOC used these FinnGen fine-mapping results and our DAP-g-based fine-mapping results for METSIM metabolites to carry out colocalization analysis assuming a single causal variant. For each FinnGen disease trait, we estimated its degree of enrichment for genome-wide associations in metabolite GWAS using TORUS[83] (https://github.com/xqwen/torus) and used this enrichment estimate as the prior for Bayesian analysis in fastENLOC. fastENLOC computes two probabilities. The regional colocalization posterior probability (RCP) is the probability of the same causal variant within a region for both the metabolite and the FinnGen disease trait. The variant colocalization posterior probability (SCP) is the probability a specific variant is causal for both traits. We limited colocalization analysis to the 1952 metabolite stepwise association signals with SPIP $\geq 0.95$ for 792 metabolites and present colocalizations for metabolite-FinnGen disease trait pairs with RCP $\geq 0.5$ (Supplementary Data 10).

**Associations of campesterol with gallstones and vanillylmandelate with hypertension in METSIM**. Among the 4698 METSIM participants with measured plasma campesterol level at baseline, we identified 199 with gallstones in METSIM (December 2020). To test for association between plasma campesterol level and presence of gallstones, we used logistic regression with covariates baseline study age, Metabolon batch, and lipid medication use. Among the 5173 METSIM participants with measured plasma vanillylmandelate level at baseline, we identified 1073 individuals with hypertension, and used logistic regression with covariates baseline study age, Metabolon batch, and hypertension medication use to test for association between plasma vanillylmandelate level and hypertension status.

**Causal effects between metabolites and FinnGen disease traits**. To infer the potential causal effects of plasma campesterol on FinnGen gallstones (phenocode: K11_CHOLELITH) and plasma vanillylmandelate on hypertension (phenocode: I9_HYPTENS), we applied four two-sample Mendelian randomization methods: inverse variance weighted[84], weighted median[85], MR-PRESSO[86], and MR-Egger[87]. These methods make different assumptions and use different strategies to account for horizontal pleiotropy, which can result in false positive inference of causality. For each metabolite, we identified nearly-independent genetic instrumental variables (LD $r^2 < 0.1$, distance $\geq 500$ kb) with unconditional single-variant association $P < 10^{-6}$. We also ran Mendelian randomization analyses to infer the causal effect of FinnGen gallstones (phenocode: K11_CHOLELITH) on campesterol and FinnGen hypertension (phenocode: I9_HYPTENS) on vanillylmandelate in a similar way. We considered findings significant if they had the same effect direction and $P < 0.05$ for all four Mendelian randomization methods. We present MR-PRESSO effect estimate and $p$-values in the main text.

**Reporting summary**. Further information on research design is available in the Nature Research Reporting Summary linked to this article.

## Data availability

NHGRI-EBI GWAS catalog: https://www.ebi.ac.uk/gwas/. Human Metabolic Individuality: http://www.metabolomix.com/list-of-all-published-gwas-with-metabolomics/. OMIM: https://omim.org/. KEGG: https://www.kegg.jp/. HMDB: https://hmdb.ca. GTEx portal: https://gtexportal.org/home/. NCBI refSeq Gene: https://www.ncbi.nlm.nih.gov/refseq/rsg/. Entrez Gene: https://www.ncbi.nlm.nih.gov/gene. UniProt: https://www.uniprot.org. Gene Ontology: http://geneontology.org. dbNSFP: https://sites.google.com/site/jpopgen/dbNSFP. FinnGen genome-wide summary statistics and Bayesian statistical fine-mapping results are available at https://r4.finngen.fi. Full summary statistics from the genome-wide association studies of the 1391 plasma metabolites are available at https://pheweb.org/metsim-metab/. METSIM individual-level data are not publicly available due to privacy restrictions on personal data. The METSIM exome sequencing and genotyping array data will be accessible through dbGaP (https://www.ncbi.nlm.nih.gov/gap/) with accession numbers phs000752 and phs000919, respectively. The METSIM WGS dataset used in this manuscript ($n = 5949$) is a subset of the full METSIM WGS data, which will be deposited into dbGaP upon completion, expected in early 2022. The METSIM metabolomics dataset ($n = 6490$) is a subset of the full METSIM metabolomics data which will be deposited into dbGaP upon completion, expected in March-May 2022. As part of data deposit in dbGAP, we will include ID lists corresponding to the individuals included in this paper. Until these data are available from dbGaP, we will provide access to the data for this paper under a Data Use Agreement to researchers who submit a short description of the proposed biomedical research project to Dr. Michael Boehnke (boehnke@umich.edu). Source data are provided with this paper.

## Code availability

Picard v2.4.1 is available at http://broadinstitute.github.io/picard/. Samtools v1.3 is available at https://github.com/samtools/. GATK v3.5 is available at https://gatk.broadinstitute.org/. VerifyBamID v1.13 is available at https://github.com/Griffan/VerifyBamID. KING v2.21 is available at https://www.kingrelatedness.com. Beagle v4.1 is

available at https://faculty.washington.edu/browning/beagle/b4_1.html. Minimac4 is available at https://github.com/statgen/Minimac4. EPACTS v3.2.6 is available at https://github.com/statgen/EPACTS. Variant Effector Predictor is available at https://useast.ensembl.org/info/docs/tools/vep/index.html. DAP-g is available at https://github.com/xqwen/dap. TORUS is available at https://github.com/xqwen/torus. fastENLOC is available at https://github.com/xqwen/fastenloc. FuzzyMatch is available at https://github.com/seamusabshere/fuzzy_match. Each use of software tools has been clearly identified in the Methods section. Integrative analysis code and scripts are available upon request from the first author.

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

## Acknowledgements

We thank the participants in the METSIM and FinnGen studies and the FinnGen study investigators. We thank Hyun Min Kang and Matthew W. Mitchell for their expertise and consultation in the genotype integration and metabolomics data processing. This work was supported by the National Institutes of Health (NIH) under awards U01 DK062370 (M.B. and A.E.L.), R35 GM138121 (X.Q.W.), R01 DK119380 (X.Q.W.), R01 GM124061 (J.Kang, E.C.H., and D.W.Z.), R01 DA048993 (J.Kang), R01 MH105561 (J.Kang), P01 HL151328 (N.O.S.), R01 HL131961 (N.O.S.), UM1 HG008853 (I.H., N.O.S., L.G., and A.E.L.), R01 DK093757 (K.L.M.), U01 DK105561 (K.L.M.), T32 HL007081 (E.Y.), and UL1 TR002345 (E.Y.). X.Y.Y. was supported by an American Diabetes Association Postdoctoral Fellowship (1-19-PDF-061) and a University of Michigan Precision Health Scholarship. M.L. was supported by the Academy of Finland (grant no. 321428) and the Sigrid Juselius Foundation. S.R. was supported by the Academy of Finland Center of Excellence in Complex Disease Genetics (grant no. 312062 and 336820), the Finnish Foundation for Cardiovascular Research, the Sigrid Juselius Foundation, University of Helsinki HiLIFE Fellow and Grand Challenge grants, and Horizon 2020 Research and Innovation Programme (grant no. 101016775 "INTER-VENE"). A.P. was supported by the Academy of Finland Center of Excellence in Complex Disease Genetics (grant no. 312074 and 336824). F.S.C., M.R.E., and L.L.B. were supported by the NIH Intramural Research Program of the National Human Genome Research Institute (ZIA HG000024).

## Author contributions

M.B., M.L., and E.B.F. supervised experiments and analyses. X.Y.Y., F.S.C., K.L.M., X.Q.W., L.J.S., E.B.F., M.L., and M.B. designed the study. A.U.J., A.E.L., C.F., H.M.S., K.T.Y., S.K.S., E.Y., L.G., I.H., I.D., M.R.E., L.L.B., N.O.S., and H.A. produced and quality-controlled the genotype and sequence data. M.L., L.F.S., and J.K. enrolled the study participants. M.L., X.Y.Y., L.F.S., J.Kang, C.F.B., and G.R.W collected, quality-controlled and/or prepared the metabolomics data for association analysis. X.Y.Y., A.U.J., H.M.S., R.W., E.B.F., L.S.C., D.B., D.W.Z., E.C.H., and J.M. analyzed data. P.V. designed and created the PheWeb site. S.R. and A.P. are principal investigators of the FinnGen study. M.L. is the principal investigator of the METSIM study. X.Y.Y., M.B., M.L., E.B.F., A.U.J., X.Q.W., and N.B.F. wrote the manuscript draft. All authors contributed to the interpretation of results and critically reviewed the manuscript.

## Competing interests

A.E.L. is an employee and stockholder of Regeneron Pharmaceuticals. L.G. is an employee of Genentech, Inc. and stockholder of Roche. N.O.S. has received research funding from Regeneron Pharmaceuticals unrelated to this work. G.R.W. is a stockholder of Metabolon, Inc. E.B.F. is an employee and stockholder of Pfizer. The remaining authors declare no competing interests.

## Additional information

## FinnGen

Samuli Ripatti ⬤ [16,17,18] & Aarno Palotie ⬤ [16,17,19]

A full list of members and their affiliations appear in the Supplementary Information file.

