## [Peer Review File · Nature Communications]

Novel signals and biological insights from GWAS of 1391 metabolites in 6136 Finnish menReviewers' Comments:

Reviewer #1:

Remarks to the Author:

The present paper describes a comprehensive genome-wide association study of blood metabolite levels (N=1,391) conducted in a relatively large sampling (N=6,136) of the Finnish population. The study has several outstanding aspects: (1) It is one of the first of its size in a population that went through a genetic bottleneck, which allows the authors to identify many novel gene-metabolite associations (mQTLs); (2) It uses exome sequencing which allows it to identify rare coding variants without the caveats that imputation brings with it; (3) It uses the latest version of the Metabolon HD4 platform, which extends the metabolome coverage beyond that of previous studies; and (4) it includes a comprehensive analysis of the X-chromosome [note that only males were studied, therefore effects of X-chromosome silencing are not an issue here]. The analysis strategy is sound and complete, the presentation of the results is clear, and the discussion is adequate. The authors share all results through a dedicated PheWeb server, which worked without problems on the tests I did. I feel that this is an important contribution to the mGWAS field and a paper of great interest for the wider readership of Nature Communications. I therefore highly recommend this paper for publication.

Reviewer #2:

Remarks to the Author:

The authors conducted an impressive amount of work, including over 1300 GWAS, systematic conditional analysis of all index signals, comparison with the existent after compiling results from over 300 existing GWAS, Mendelian Randomization analyses, SNP-to-gene mapping, investigation of the links between metabolite associations and disease association, and finally a focus on a few interesting associations. I do not have any major concern with the methods used, which are all pretty well established. The Web platform for sharing the result is also a nice addition to the work. My major and minor comments are mostly about clarifying a few points.

Major

1. There are many, many, results in this study. After a few readings, it was a bit unclear what was the key message. I found the highlight on population genetics (increased number of rare causal variants in Finnish population...) in the abstract to be very helpful in this regard. Unfortunately, the bulk of the result is flatter. Any updates/reorganization of the text conveying that key message along the results would benefit this otherwise amazing work.
2. Line 163. "In each of the 1,952 credible sets, we identified the variant with the largest VPIP. This list comprised 1,119 distinct variants, 100 with MAF>10-fold greater in METSIM than in NFE." Assuming there is at least one variant per credible set, there should be one "largest VPIP" variant per set. Why is there 1,119 variants –pleiotropy?
3. Line 302-311. The colocalization analyses merged the metabolite GWAS and FinnGen disease GWAS. What was the SNP overlap between FinnGen and METSIM? In particular, what was the coverage for the many rare variants in METSIM used in the present analysis. If only partial, is there a risk that some likely causal variants are discarded, and that some of the colocalization only point to (more common) tagging SNPs?
4. Line 335-338. "Mendelian randomization analysis using 15 independent variants for campesterol suggested a causal effect of lower plasma campesterol level on higher gallstone risk (Methods; $\beta = -0.70$, $P = 7.2 \times 10^{-8}$; Fig. 4b-c). Interestingly, we also identified a causal effect of gallstones on lower plasma campesterol level ($\beta = -0.49$, $P = 1.3 \times 10^{-39}$) in Mendelian randomization analysis using 151 independent variants." If I understand correctly, there is two opposite causal effects (campesterol \rightarrow gallstone risk, and gallstone risk \rightarrow campesterol). Or am I missing something? If no, this is weird. By definition, causal effects are expected to be unidirectional. Some sort of a causal loop is not impossible but seems quite unlikely, and sounds more like a red flag on the validity of the MR. The author should be very cautious with the interpretation of this result, and at least 1) highlight that this is an

unexpected result, and 2) remind the reader on the limitations of MR in general.

5. Line 354-367. The hypertension/vanillylmandelate analysis is interesting, but –despite the density of the manuscript– deserves a few additional analyses: 1) it would be of interest to have the R^2 (i.e. the squared correlation) between hypertension and vanillylmandelate on top of the p-val (“ $\beta=0.52$, $P=3.2 \times 10^{-9}$ ”). If R^2 is large, it is unlikely to be explained by the DBH p.Arg79Trp variant alone, suggesting other factors are involved. 2) regarding the hypothesis of DBH being involved in two distinct pathways, it would be of interest to conduct adjusted analyses (DBH-vanillylmandelate test adjusted for hypertension, and conversely), and check how the changes in estimates match that hypothesis. 3) The supp Fig 11 seems to indicate a modest trend, and there is only 15 SNPs for the MR using vanillylmandelate as the causal factor. Is there a possible power issue?

Minor:

6. It is quite challenging to synthesize the count of signals. I’d suggest to move supp Fig1 to the main text.

7. The fine mapping uses the concept of credible set. While a reference is cited and a description provided in the method, it would be helpful for the reader if the term was broadly re-defined in the main text (the minimal subset of variants for which the sum of posterior probability is ≥ 0.95 ?).

8. A double transformation is performed on all metabolites (rank inverse normal, then adjustment, and rank inverse normal on the residual). I am wondering how much these transformations impact the association signal. To be clear, I am not challenging the validity of the test (and supp Fig 2 is quite convincing about that), but rather asking about the impact of those transformations on power. A sensitivity analysis (with e.g. a single transformation or no transformation) of a few top associations can easily answer that question.

9. It seems that the GWAS was conducted using the EMMAX model implemented in EPACTS. Two questions: 1) the authors mentioned they excluded closely related individuals (“estimated kinship ≥ 0.125 ”), what was the argument for using an LMM? there was still substantial relatedness? and 2) EMMAX is not known to be very fast. How much computational time did it take to run over 1,000 GWAS each including over 10M SNPs? And why not using more recent implementation of LMM (e.g. BoltLMM, PMID=25642633). This might be helpful to other researcher conducting large-scale analyses.

10. Several recent papers have investigated the relationship between MAF and effect size using the alpha model (e.g. PMID=30770844, 28530675). It would be quite easy and of great interest to the community to derive an empirical estimate of alpha using data from supp Fig 4.

11. Based on supp Fig 13, it seems that ~a quarter of metabolites has missing data over 30%. Imputation was conducted using a KNN approach. I wonder whether imputation might explain part of the observed pleiotropy. The authors might conduct a quick sanity check, deriving e.g. the correlation between N before Imputation and a dummy variable indicating whether the metabolite share associated variants with other metabolites.

Genome-wide association study of 1,391 plasma metabolites in 6,136 Finnish men identifies 303 novel signals and provides biological insights into human diseases (NCOMMS-21-32099-T)

**Point-by-Point Responses
October 27, 2021**

Reviewer #1:

The present paper describes a comprehensive genome-wide association study of blood metabolite levels (N=1,391) conducted in a relatively large sampling (N=6,136) of the Finnish population. The study has several outstanding aspects: (1) It is one of the first of its size in a population that went through a genetic bottleneck, which allows the authors to identify many novel gene-metabolite associations (mQTLs); (2) It uses exome sequencing which allows it to identify rare coding variants without the caveats that imputation brings with it; (3) It uses the latest version of the Metabolon HD4 platform, which extends the metabolome coverage beyond that of previous studies; and (4) it includes a comprehensive analysis of the X-chromosome [note that only males were studied, therefore effects of X-chromosome silencing are not an issue here]. The analysis strategy is sound and complete, the presentation of the results is clear, and the discussion is adequate. The authors share all results through a dedicated PheWeb server, which worked without problems on the tests I did. I feel that this is an important contribution to the mGWAS field and a paper of great interest for the wider readership of Nature Communications. I therefore highly recommend this paper for publication.

Response: We thank the reviewer for the generous comments.

Reviewer #2:

The authors conducted an impressive amount of work, including over 1300 GWAS, systematic conditional analysis of all index signals, comparison with the existent after compiling results from over 300 existing GWAS, Mendelian Randomization analyses, SNP-to-gene mapping, investigation of the links between metabolite associations and disease association, and finally a focus on a few interesting associations. I do not have any major concern with the methods used, which are all pretty well established. The Web platform for sharing the result is also a nice addition to the work. My major and minor comments are mostly about clarifying a few points.

Response: We thank the reviewer for the positive comments.

Major

1. There are many, many, results in this study. After a few readings, it was a bit unclear what was the key message. I found the highlight on population genetics (increased number of rare causal variants in Finnish population...) in the abstract to be very helpful in this regard. Unfortunately, the bulk of the result is flatter. Any updates/reorganization of the text conveying that key message along the results would benefit this otherwise amazing work.

Response: We thank the reviewer for this helpful comment. The utility of Finnish population for genetic discovery is our key message. We used the current organization to convey that

message and to highlight what we considered the most interesting results from the many analyses undertaken.

To better address the organization issue, we change Supplementary Figure 1 to main Figure 1, as the reviewer suggested in comment 6 (below). This figure outlines our analyses and main findings from each analysis, and so helps understand the overall organization of the paper. We also include a new section at the beginning of the Results to help set the stage for what follows:

“Study design. We assayed 1,544 plasma metabolites using the Metabolon DiscoveryHD4 mass spectrometry platform (Supplementary Tables 1-2) in 6,136 randomly-selected METSIM participants who were non-diabetic at baseline and passed quality control (QC) (Supplementary Table 3; Fig. 1). 1,391 metabolites were successfully quantified in ≥ 500 of these 6,136 participants. We created a METSIM imputation reference panel of $>26M$ genetic variants by integrating genome and exome sequence and array genotypes in 2,922 METSIM participants (Methods; Supplementary Table 4). We used this reference panel to impute genotypes in all METSIM participants. To discover genetic mechanisms for plasma metabolite levels, we performed GWAS and statistical fine-mapping analysis and nominated putative causal genes for metabolites. We integrated metabolomics with FinnGen disease GWAS to understand disease mechanisms through genetic colocalization and Mendelian randomization analysis (Fig. 1).

2. Line 163. “In each of the 1,952 credible sets, we identified the variant with the largest VPIP. This list comprised 1,119 distinct variants, 100 with MAF >10 -fold greater in METSIM than in NFE.” Assuming there is at least one variant per credible set, there should be one “largest VPIP” variant per set. Why is there 1,119 variants –pleiotropy?

Response: The reviewer is correct that pleiotropy is responsible for this difference. Among the variants with largest VPIP for the 1,952 credible sets, 823 variants were identified in a single credible set, 150 in two credible sets, 56 in three, 23 in four, 21 in five, 14 in six, 11 in seven, 5 in eight, 1 in nine, 1 in 10, 3 in 12, 1 in 13, 1 in 14, 2 in 15, 1 in 16, 2 in 17, 1 in 18, 1 in 19, 1 in 25, and 1 in 39. We made this pleiotropy clear in the revised manuscript on page 9 as:

“Among the 1,119 variants, 150 were shared between two signals and 146 by ≥ 3 (up to 39).”

3. Line 302-311. The colocalization analyses merged the metabolite GWAS and FinnGen disease GWAS. What was the SNP overlap between FinnGEN and METSIM? In particular, what was the coverage for the many rare variants in METSIM used in the present analysis. If only partial, is there a risk that some likely causal variants are discarded, and that some of the colocalization only point to (more common) tagging SNPs?

Response: We analyzed up to 16,172,108 genetic variants with minor allele count (MAC) ≥ 5 for metabolites in METSIM, while FinnGen release 4 analyzed up to 16,381,641 variants with MAC ≥ 5 . For all the 16,172,108 METSIM variants, 83.8% are available in FinnGen. For the 6,562,945 rare variants we analyzed in METSIM (MAC ≥ 5 , minor allele frequency $<1\%$), 75.3% are available in FinnGen.

The reviewer is correct. Despite the proportion of overlapping variants being relatively high between METSIM and FinnGen in our study, we could have missed colocalizations when the true causal variants are discarded in one dataset and the most likely result is that we detected colocalizations at more common variants. We thank the reviewer for pointing out this limitation which we now address on page 16 by stating: *“We may have missed colocalizations where the*

true causal variants were discarded in METSIM or FinnGen; in such cases, we may have detected colocalizations at other, likely more common, variants.”

4. Line 335-338. “Mendelian randomization analysis using 15 independent variants for campesterol suggested a causal effect of lower plasma campesterol level on higher gallstone risk (Methods; $\beta=-0.70$, $P=7.2\times 10^{-8}$; Fig. 4b-c). Interestingly, we also identified a causal effect of gallstones on lower plasma campesterol level ($\beta=-0.49$, $P=1.3\times 10^{-39}$) in Mendelian randomization analysis using 151 independent variants.” If I understand correctly, there is two opposite causal effects (campesterol \rightarrow gallstone risk, and gallstone risk \rightarrow campesterol). Or am I missing something? If no, this is weird. By definition, causal effects are expected to be unidirectional. Some sort of a causal loop is not impossible but seems quite unlikely, and sounds more like a red flag on the validity of the MR. The author should be very cautious with the interpretation of this result, and at least 1) highlight that this is an unexpected result, and 2) remind the reader on the limitations of MR in general.

Response: We thank the reviewer for this comment. We agree with the reviewer that the bidirectional causal relationship could exist but requires additionally careful considerations. For the causal findings between campesterol and gallstones specifically, we applied the Steiger method (PMID: 29149188) that is designed to test for whether the assumption of causal directionality from exposure to outcome is valid in the Mendelian randomization analysis from campesterol to gallstones and vice versa. The Steiger analysis suggested that the causality of campesterol on gallstones is a valid assumption (causal direction=true, $P=6.9\times 10^{-75}$) but the assumption that gallstones causes plasma campesterol level is not (causal direction=false, $P=6.1\times 10^{-72}$). The causal effect of gallstones on campesterol that we previously found could be induced by invalid instrument variables, which fulfill the p -value threshold because of the large sample size in the FinnGen study, but confer larger effect on outcome in comparison with exposure (PMID: 31298278). The inclusion of these likely invalid instrument variables violates the “exclusion restriction” assumption in Mendelian randomization analysis, which would cause identification of a spurious causal relationship.

Based on this analysis, we removed the Mendelian randomization result from gallstones to campesterol so that the revised paragraph on page 17 now reads:

“Mendelian randomization analysis using 15 independent variants for campesterol suggested a causal effect of lower plasma campesterol level on higher gallstone risk (Methods; $\beta=-0.70$, $P=7.2\times 10^{-8}$; Fig. 5b-c). ABCG5 and ABCG8 together encode a heterodimeric ATP-binding cassette transporter that facilitates secretion of cholesterol and non-cholesterol sterols in the intestine and bile. High plasma campesterol levels might compete with cholesterol for ABCG5/ABCG8 transporters during biliary cholesterol secretion, resulting in decreased biliary cholesterol levels and reduced risk of gallstones (Supplementary Fig. 10).”

5. Line 354-367. The hypertension/vanillylmandelate analysis is interesting, but –despite the density of the manuscript– deserves a few additional analyses: 1) it would be of interest to have the R^2 (i.e. the squared correlation) between hypertension and vanillylmandelate on top of the p -val (“ $\beta=0.52$, $P=3.2\times 10^{-9}$ ”). If R^2 is large, it is unlikely to be explained by the DBH p.Arg79Trp variant alone, suggesting other factors are involved. 2) regarding the hypothesis of DBH being involved in two distinct pathways, it would be of interest to conduct adjusted analyses (DBH-vanillylmandelate test adjusted for hypertension, and conversely), and check how the changes in estimates match that hypothesis. 3) The supp Fig 11 seems to indicate a modest trend, and there is only 15

SNPs for the MR using vanillylmandelate as the causal factor. Is there a possible power issue?

Response: The squared correlation between vanillylmandelate and hypertension in METSIM is very low: $R^2=0.01$. We changed the relevant sentence by adding the R^2 estimation on page 18 in the revised manuscript: *“In 5,173 METSIM participants with observed vanillylmandelate levels, 1,073 with hypertension, plasma vanillylmandelate level was weakly but significantly associated with hypertension ($\beta=0.52$, $R^2=0.01$, $P=3.2\times 10^{-9}$).”*

We re-ran genetic association in METSIM for the lead variant p.Arg79Trp (rs77273740) with vanillylmandelate after controlling for hypertension status. Results with and without adjustment were nearly identical:

Model	β	SE	P-value
Without hypertension adjustment	-0.3829	0.0480	1.79×10^{-15}
With hypertension adjustment	-0.3825	0.0479	1.63×10^{-15}

We included this result by adding one sentence on page 18 in the revised manuscript as follow: *“Controlling for hypertension status gave nearly identical genetic association between DBH p.Arg79Trp and vanillylmandelate ($\beta=-0.38$, $P=1.6\times 10^{-15}$).”*

We agree with the reviewer that we cannot rule out the possibility of insufficient statistical power being responsible for the lack of significance for a causal relationship between vanillylmandelate and hypertension in the Mendelian randomization analysis. To acknowledge that, we modified the relevant sentence on page 18:

“Using Mendelian randomization, we did not find significant evidence of causal effects of vanilylmandelate on hypertension ($P=0.15$; 10 independent variants; Supplementary Fig. 10) or of hypertension on vanillylmandelate ($P=0.17$; 157 independent variants), suggesting this signal conferred effects on hypertension risk and vanillylmandelate through different pathways, consistent with the two possible DBH effects identified by our knowledge-based approach (Supplementary Fig. 10). The analysis from vanillylmandelate to hypertension could only make use of ten instruments and so may be underpowered.”

Minor:

6. It is quite challenging to synthesize the count of signals. I'd suggest to move supp Fig1 to the main text.

Response: We thank the reviewer for this helpful suggestion. Done, as noted above.

7. The fine mapping uses the concept of credible set. While a reference is cited and a description provided in the method, it would be helpful for the reader if the term was broadly re-defined in the main text (the minimal subset of variants for which the sum of posterior probability is ≥ 0.95 ?).

Response: We agree. On page 9, we added a definition of credible set:

“For these 1,952 signals, we built 95% credible sets of potential causal variants, the minimal subset of variants with summed VPIP ≥ 0.95 ”.

8. A double transformation is performed on all metabolites (rank inverse normal, then adjustment, and rank inverse normal on the residual). I am wondering how much these transformations impact the association signal. To be clear, I am not challenging the validity of the test (and supp Fig 2 is quite convincing about that), but rather asking about the impact of those transformations on power. A sensitivity analysis (with e.g. a single transformation or no transformation) of a few top associations can easily answer that question.

Response: We thank the reviewer for this suggestion. To evaluate the impact of double trait transformation on the genetic association analysis, we randomly selected 50 of the 2,030 association signals and re-ran single-variant association tests using the single transformed metabolomics data. The scatter plots of p -values and effect sizes for these 50 genetic associations using double and single trait transformation suggest trivial impact of double trait transformation on the power of genetic associations.

9. It seems that the GWAS was conducted using the EMMAX model implemented in EPACTS. Two questions: 1) the authors mentioned they excluded closely related individuals (“estimated kinship ≥ 0.125 ”), what was the argument for using an LMM? there was still substantial relatedness? and 2) EMMAX is not known to be very fast. How much computational time did it take to run over 1,000 GWAS each including over 10M SNPs? And why not using more recent implementation of LMM (e.g. BoltLMM, PMID=25642633). This might be helpful to other researcher conducting large-scale analyses.

Response: We excluded close relatives (on page 25 in Method section of METSIM integrative panel and genotype imputation) when we created reference panel for genotype imputation, but did not exclude relatives in the downstream analyses including GWAS. We have clarified this issue by revising the relevant sentence in the Methods section on page 27 as follow:

“To account for sample relatedness and potential population stratification among the 6,136 participants in our analysis set, we carried out single-variant association tests using a linear mixed model in EPACTS (v3.2.6) (<https://github.com/statgen/EPACTS>) on the normalized residual metabolite values.”

We acknowledge that EMMAX is not the most efficient tool for such large-scale GWAS now. However, we began our analyses for the paper several years ago and there were fewer options at that time. We did consider using BOLT-LMM which is more computationally efficient than EMMAX. However, the algorithms used in BOLT-LMM rely on approximations that require relatively large sample sizes; the developer recommends $n > 5,000$. In our study, sample sizes for 482 of the 1,391 metabolites were less than 5,000. We felt it better to use a single software platform for all analyses.

10. Several recent papers have investigated the relationship between MAF and effect size using the alpha model (e.g. PMID=30770844, 28530675). It would be quite easy and of great interest to the community to derive an empirical estimate of alpha using data from supp Fig 4.

Response: We thank the reviewer for suggesting this interesting alpha model. Using the approach described in PMID: 30770844, we built minor allele frequency-dependent genetic relatedness matrix (GRM) and estimated the alpha value for each of the 1,391 metabolites that we investigated in this study. We found 829 of the 1,391 metabolites have an alpha above -0.0001, 561 below -0.9, and 1 at -0.24. We realized that these alpha estimates relied on the profile likelihoods that were estimated in the restricted maximum likelihood (REML) algorithm in GCTA. Unsurprisingly, the likelihood ratio test was not significant for 1,070 metabolites at $P < 0.05$ and 1,354 at $P < 0.05/1,391$ in GCTA REML probably because of the relatively small sample size. As a result, these alpha estimates might be misleading and we preferred not to include them in the revised manuscript. We hope we could pursue this alpha approach in the near future when we have larger study sample size.

11. Based on supp Fig 13, it seems that ~a quarter of metabolites has missing data over 30%. Imputation was conducted using a KNN approach. I wonder whether imputation might explain part of the observed pleiotropy. The authors might conduct a quick sanity check, deriving e.g. the correlation between N before Imputation and a dummy variable indicating whether the metabolite share associated variants with other metabolites.

Response: We imputed missing metabolomics data using a KNN approach when we performed principal component analysis to identify the number of principal components that explain at least 95% phenotypic variance among the 1,391 metabolites. However, we did not impute the missing metabolomics data in our GWAS. To clarify this point, we modified the relevant sentence in the Methods section on page 27 as follow:

“We limited analysis to the 1,391 metabolites successfully measured on ≥ 500 METSIM participants and to the genetic variants with minor allele count (MAC) ≥ 5 ; we did not impute missing metabolite data.”

Reviewers' Comments:

Reviewer #2:

Remarks to the Author:

The authors addressed all of my comments, and I have no further concerns with the manuscript.